# Physicochemical and microbiome changes in queso Crema de Chiapas during ripening

**Blanca Nayelli Ocampo Morales[1], Arturo Hernández Montes[1], Karel Estrada[2]\*,
Ernestina Valadez Moctezuma[3]\***

**1** Departamento de Ingeniería Agroindustrial, Universidad Autónoma Chapingo, Texcoco, Estado de México, México, **2** Instituto de Biotecnología, Universidad Nacional Autónoma de México, Chamilpa, Morelos, México, **3** Departamento de Fitotecnia, Universidad Autónoma Chapingo, Texcoco, Estado de México, México

\* karel.estrada@ibt.unam.mx (KE); evaladezm@chapingo.mx (EVM)

## Abstract

The dynamic changes in the physicochemical, microbiological, and metagenomic profiles of Crema de Chiapas cheese were evaluated across three ripening stages (2, 29, and 58 days). Although the main physicochemical properties —including fat content— remained remarkably stable, salt and protein levels showed noticeable variation throughout ripening. Protein content had the strongest influence on sample differentiation across ripening stages in unsupervised multivariate models, enabling the clustering of microbial diversity according to maturation time. A clear shift in microbial diversity was detected, marked by a reduction in bacterial genera and a concurrent decline in fungal and yeast populations as ripening advanced. The predominant bacterial genera throughout ripening were Streptococcus, Lactobacillus, and Lactococcus. While Streptococcus and Lactobacillus increased over time, Lactococcus exhibited the opposite trend. Metagenomic analysis revealed a decrease in Candida etchellsii and a concomitant increase in Candida tropicalis as ripening progressed. Quantitative PCR (qPCR) confirmed the presence of C. etchellsii at T1 (Ct = 7.22) and C. tropicalis at T3 (Ct = 9.84). The presence of three additional bacterial genera—Chryseobacterium, Aeromonas, and Enterobacter—identified by next-generation sequencing (NGS), was also assessed by qPCR. Chryseobacterium was detected at T2 (Ct = 3.26), whereas Aeromonas and Enterobacter were absent across all stages. Collectively, these findings suggest that potentially pathogenic microorganisms were not present at biologically relevant levels.

## Introduction

A variety of artisanal cheeses are produced in Mexico, including Queso Crema de Chiapas, whose production is characterized by resting raw milk, prolonged paste fermentation, and kneading [1]. As ripening proceeds, the cheese develops textures ranging

**Data availability statement:** All sequencing data have been deposited in the NCBI BioProject database under accession number PRJNA1247530, with SRA experiment accessions SRX28272105-SRX28272110. https://www.ncbi.nlm.nih.gov/bioproject/PRJNA1247530/.

**Funding:** Consejo Nacional de Humanidades Ciencias y Tecnologías (CONAHCYT) Grant 2020-000013-01NACF-03858. The funders had no role in study design, data collection and analysis, decision to publish, or preparation of the manuscript.

**Competing interests:** The authors have declared that no competing interests exist.

from soft to semi-hard or hard [2]. It is typically sold in rectangular blocks wrapped in plastic, aluminum foil, or cellophane. Crema de Chiapas may be marketed as a plain or seasoned paste and is consumed from one week after production up to more than three months of ripening [3]. This cheese has been reported to contain bioactive compounds with antioxidant, angiotensin-converting enzyme (ACE) inhibitory, and antimicrobial activities [4–6]. Lactic acid bacteria (LAB) and fungi contribute to desirable sensory characteristics, including texture, aroma, and flavor [7]. However, peptide hydrolysis by bacteria can also yield sour, sweet, or undesirable off-flavors. Through protein and fat degradation, fungi further generate diverse volatile compounds [8]. The microbial composition of cheese is influenced by physicochemical parameters such as ripening time and salt concentration [9], as well as by milking conditions, herd diet, manufacturing procedures, storage environments, and complex microorganism–component and microorganism–microorganism interactions [10].

Metagenomic approaches based on 16S rRNA gene sequencing for bacteria and internal transcribed spacer (ITS) sequencing for fungi and yeasts have enabled comprehensive profiling of microbial communities in cheese. The 16S rRNA gene, located in the small ribosomal subunit, is approximately 1,500 bp long and contains nine hypervariable regions [11,12], among which V3–V4 are commonly used for bacterial identification in cheese [13,14]. The ITS region, approximately 600 bp in length, comprises two spacers (ITS1 and ITS2) and allows the identification of fungal and yeast species in cheeses and other fermented foods [15–18]. In this study, Queso Crema de Chiapas from Mexico was characterized through physicochemical and microbiological analyses, combined with metagenomic identification of microorganisms across different ripening stages.

## Materials and methods

### Sampling of Queso Crema de Chiapas

Cheeses were produced from raw bovine milk, and three samples were collected from each of three batches manufactured on consecutive days. The cheeses were allowed to mature for distinct durations: T1 = 2 days, T2 = 29 days, and T3 = 58 days, with three replicates per condition (n = 3). Samples were obtained from a cheese factory located in Pijijiapan, Chiapas, Mexico (15°43′39.9354″ N, 93°15′45.4314″ W). To preserve the native microbial dynamics at each ripening stage, samples were transported to the laboratory in an insulated container without active cooling, thus preventing drastic temperature shifts. For mature cheeses (T2 and T3), the low water activity and pH of the matrix ensured microbial community stability, while for fresh cheese (T1), the lack of cooling prevented cold-shock effects that could alter early microbial metabolism.

### Physicochemical analysis of Queso Crema de Chiapas

Physicochemical parameters—including protein, fat, total solids, and NaCl—were determined using a FoodScan™ Lab equipment (FOSS Analytical AB, Hillerød, Denmark). Moisture [19] and ash [20] contents were quantified by gravimetric methods.

Each measurement was performed in triplicate. Water activity ($a_w$) was determined using an Aqualab Series 3 instrument (Decagon Devices Inc., Washington, USA) [21].

Carbohydrate (CHO´S) content was estimated by difference: [CHO'S (%) = Total Solids (TS) %-(fat % + protein % + ash %)] [22].

## Microbiological analysis

For microbiological analyses, six blocks of cheese (n = 6) from three different batches were used, with two replicates each. A 10g portion was taken from the interior of each block [23] and homogenized with 90 mL of sterile 0.1% peptone water [24]. Aerobic mesophilic bacteria counts were performed according to NOM-092-SSA1–1994. For fungi and yeasts, NOM-111-SSA1–1994 was followed, and total coliform bacteria were quantified using NOM-113-SSA1–1994.

## Identification of Bacteria, fungi, and yeasts by metagenomics

**DNA extraction.** DNA was extracted from Queso Crema de Chiapas samples collected at the three ripening stages, with two replicates per stage. Replicates from the same stage were pooled prior to extraction. Samples were ground in liquid nitrogen until very fine particles were obtained. Then, 0.1g of cheese was weighed, and DNA was extracted following the protocol of the ZymoBIOMICS DNA kit (California, USA). DNA was then quantified with a UVS-99 NanoDrop (Avans Biotechnology, Taipei, Taiwan, China) at 260 nm to assess purity and integrity.

**Sequencing.** The V3–V4 regions of the 16S rRNA gene and the internal transcribed spacers (ITS1–ITS2) were sequenced on an Illumina MiSeq platform to identify bacterial and fungal/yeast communities, respectively. All sequencing data have been deposited in the NCBI BioProject database under accession number PRJNA1247530, with SRA experiment accessions SRX28272105–SRX28272110.

## Real-time quantitative PCR (qPCR)

A two-step real-time quantitative PCR assay was performed to determine the presence of selected bacterial genera and yeast species identified through metagenomic analysis in Crema de Chiapas cheese across the three ripening stages. Primers were specifically designed for each microorganism using the Primer3 software [25], and their characteristics are listed in Table 1. qPCR reactions were conducted using the SYBR® Green Supermix kit (BIO-RAD, California, USA) according to the manufacturer's instructions, in a CFX96 Deep Well thermocycler (BIO-RAD, California, USA). Thermocycling conditions included an initial step at 98 °C for 2 min, followed by 39 cycles of 98 °C for 2 s (denaturation) and 60 °C for 20 s (annealing/extension).

Table 1. Primers for bacterial and yeast detection.

| Microorganism | Primer | Primer size(bp) | Amplicon size (bp) |
|---|---|---|---|
| *Aeromonas* | L: 5´-AGATGTGAAAGCCCCGGG-3´ | 18 | 185 |
| | R: 5'-TGTTTGCTCCCCACGCT-3´ | 17 | |
| *Enterobacter* | L:5´-GAATACCGGTGGCGAAGG-3´ | 18 | 188 |
| | R: 5´-TTGCGGCCGTACTCCC-3´ | 16 | |
| *Chryseobacterium* | L: 5´-GCGTGGGGAGCGAACA-3' | 16 | 215 |
| | R: 5'-TGGTAAGGTTCCTCGCGT-3' | 18 | |
| *Candida etchellsii* | L: 5´-GGAGCGCCGAACTTTCTC-3´ | 18 | 156 |
| | R: 5´-TGCTTAAGTTCGGCGGGT-3´ | 18 | |
| *Candida tropicalis* | L: 5´-GGAGCAATCCTACCGCCA-3´ | 18 | 234 |
| | R: 5´-ACGCTCAAACAGGCATGC-3´ | 18 | |

GAPDH was used as an internal positive control at all ripening stages, along with an external positive control (GAPDH RNA) and a negative control containing water instead of DNA. GAPDH is commonly used in qPCR due to its stability and because its expression is generally unaffected by different experimental conditions [26]. For each microorganism, a melt-curve analysis was performed using the preset increments and temperature profiles provided by the instrument.

### Statistical analysis

**Physicochemical and microbiological statistical analysis.** The Kolmogorov–Smirnov normality test was applied using the MINITAB 2017 (Minitab, LLC, Pine Hall Road, PA, USA) to evaluate the variables $a_w$, moisture, protein, fat, ash, salt, TS, CHO's, mesophilic counts, and fungal/yeast counts across the three ripening stages. Variables that satisfied the assumption of normality were analyzed using a Completely Randomized Design (CRD) with $\alpha = 0.05$. Mean comparisons were performed with the Tukey–Kramer test ($\alpha = 0.05$) in MINITAB 2017. When the data did not meet, the Kruskal–Wallis test was applied [27]. In these cases, mean comparisons were carried out using the Steel–Dwass–Critchlow–Fligner test with $\alpha = 0.05$, and analyses were performed using XLSTAT 2019 (XLSTAT, LLC, Rue Damrémont, Paris, France).

Principal component analysis (PCA) was performed using PAST v.5.2.1 (Hammer et al., Oslo, Norway), and samples were subsequently clustered by the K-means algorithm. Three-dimensional visualizations of sample distributions were generated using Plotly Chart Studio (Plotly Technologies Inc., Montreal, Canada). Permutational multivariate analysis of variance (PERMANOVA) was carried out in PAST v.5.2.1, complemented by principal coordinate analysis (PCoA) based on the Bray–Curtis dissimilarity metric.

**Bioinformatics analysis of metagenomics.** Quality control of the 16S rRNA gene sequences and ITS regions was performed using the MultiFastQC program on the Galaxy platform [28]. Adapter removal and subsequent sequence quality checks were carried out using the Trimmomatic and MultiFastQC programs, respectively. Flash v1.2.11 (https://ccb.jhu.edu/software/FL-

ASH/) was used to merge paired-end amplicon sequences; duplicate and chimeric sequences were subsequently removed using the Vsearch program (https://github.com/torognes/vsearch). Taxonomic profiling was conducted using Parallel-Meta Suite version 3.7 (http://bioinfo.single-cell.cn/parallel_meta.html), with the Greengenes 16S database (May 2013) for bacterial identification. Taxonomic assignment of ITS sequences was performed using the UNITE database v8.2 (release: 2020-02-04), which incorporates dynamic Species Hypotheses. Amplicon Sequence Variants (ASVs) were defined at a 99% sequence similarity threshold, and assignments were retained only if they met a minimum confidence level of 0.80.

Statistical and ecological analyses were carried out within the R environment (v4.2.2). The abundance table, taxonomic classifications, and sample metadata were first integrated into a unified object using the phyloseq package. This object served as the basis for all downstream analyses, including the calculation of alpha diversity metrics (Shannon, Simpson, and Chao1 indices). Prior to alpha diversity analysis, the ASV abundance table was normalized by rarefaction using the rarefy_even_depth function of the phyloseq package, and a fixed random seed was applied to ensure full reproducibility. Finally, all visualizations—including Venn diagrams, stacked bar plots of amplicon relative abundance, and the integrated heatmap—were generated using R and Perl scripts.

## Results and discussion

### Physicochemical analysis of Queso Crema de Chiapas

Table 2 shows that all variables followed a normal distribution across the three maturation times, except for fungi and yeasts at T3.

The results of the statistical analyses are summarized in Table 3. Moisture content ranged from 33.36% to 34.56%, and fat content from 35.15% to 36.9%; neither parameter showed significant differences during ripening.

**Table 2. Kolmogorov–Smirnov normality test probability values by variable.**

| Variables | Time | KS | P | Normality |
|---|---|---|---|---|
| $a_w$ | T1 | 0.29 | 0.15 | Yes |
| | T2 | 0.39 | 0.08 | Yes |
| | T3 | 0.25 | 0.15 | Yes |
| Moisture | T1 | 0.31 | 0.15 | Yes |
| | T2 | 0.32 | 0.15 | Yes |
| | T3 | 0.20 | 0.15 | Yes |
| Protein | T1 | 0.27 | 0.15 | Yes |
| | T2 | 0.20 | 0.15 | Yes |
| | T3 | 0.36 | 0.13 | Yes |
| Fat | T1 | 0.20 | 0.15 | Yes |
| | T2 | 0.37 | 0.11 | Yes |
| | T3 | 0.37 | 0.10 | Yes |
| Ash | T1 | 0.39 | 0.08 | Yes |
| | T2 | 0.37 | 0.10 | Yes |
| | T3 | 0.33 | 0.15 | Yes |
| Salt | T1 | 0.38 | 0.09 | Yes |
| | T2 | 0.37 | 0.10 | Yes |
| | T3 | 0.18 | 0.15 | Yes |
| ST | T1 | 0.28 | 0.15 | Yes |
| | T2 | 0.19 | 0.15 | Yes |
| | T3 | 0.19 | 0.15 | Yes |
| CHO´S | T1 | 0.19 | 0.15 | Yes |
| | T2 | 0.20 | 0.15 | Yes |
| | T3 | 0.31 | 0.15 | Yes |
| Mesophiles | T1 | 0.26 | 0.15 | Yes |
| | T2 | 0.26 | 0.15 | Yes |
| | T3 | 0.26 | 0.15 | Yes |
| Fungi and yeasts | T1 | 0.26 | 0.15 | Yes |
| | T2 | 0.26 | 0.15 | Yes |
| | T3 | – | – | No |

A p-value greater than 0.05 indicates that the data follow a normal distribution.

This behavior aligns with that reported for vacuum-packed San Simón da Costa cheese [29], where packaging prevented moisture loss and the associated decline in $a_w$ and fat content. Fat content typically remains stable because of the limited lipolytic activity of lactic acid bacteria (LAB), as both saturated and unsaturated fatty acids exhibit toxic effects on these microorganisms [30]. Moreover, the optimal pH for lipolytic enzyme activity (~9) is not reached during cheese ripening [31].

Among the variables measured, reductions were observed in total solids (TS), protein, and salt contents (Table 3). TS decreased from 69.26% to 64.61%, protein from 27.81% to 22.87%, and salt from 2.73% to 1.91%. Similar decreases in total solids and protein during ripening have been reported for Örgü cheese [32] and Greek Feta cheese [33], attributed to progressive degradation reactions that reduce these components over time. In both Feta and artisanal raw sheep milk cheeses, declines in salt concentration are linked to elevated pH and fat levels, which can limit salt diffusion and absorption [34,35].

**Table 3. Physicochemical analysis of Queso Crema de Chiapas at different ripening times.**

| Variables | T1 | T2 | T3 |
|---|---|---|---|
| $a_w$ | 0.93±0.02a | 0.93±0.02a | 0.93±0.02a |
| Moisture (%) | 34.56±0.90a | 33.36±2.47a | 33.97±0.39a |
| Protein (%) | 27.81±1.00a | 25.70±0.39b | 22.87±0.78c |
| Fat (%) | 36.01±0.52a | 35.13±2.42a | 36.90±0.85a |
| Ash (%) | 3.54±0.89a | 3.35±0.75a | 3.58±0.08a |
| Salt (%) | 2.73±0.23a | 1.98±0.11b | 1.91±0.29b |
| Total solids (%) | 69.26±1.90a | 64.61±1.51b | 64.61±1.51b |
| Carbohydrates (%) | 1.91±1.27a | 0.43±0.36a | 1.27±0.97a |

## Microbiological analysis of Queso Crema de Chiapas

Across all ripening stages (T1 = 2 d, T2 = 29 d, T3 = 58 d), no growth of total coliforms was detected. This absence can be attributed to the production of antimicrobial compounds—such as organic acids and ethanol—during ripening, as well as to physicochemical changes in the cheese matrix, including reductions in pH, $a_w$, and Moisture [36], [37].

Mesophilic bacteria counts did not show significant statistical differences across the three ripening stages, ranging from 4.36 to 6.13 $\log_{10}$ CFU $g^{-1}$ (Fig 1). This consistency may reflect the limited changes in the cheese's physicochemical properties during ripening. Fungal and yeast counts similarly showed no significant variation across stages, likely due to minimal physicochemical fluctuations (Fig 1). Some studies report that the presence of fungi in cheese is controversial, as they are airborne and certain species can produce toxins [38], [39]. In industrial Oaxaca cheese, for instance, aflatoxin B1 and aflatoxicol have reportedly been detected, originating from corn flour added to the cheese dough as a thickening agent [40]. However, it is well established that specific genera of fungi and yeasts play an important role in cheese production, as they contribute to proteolysis and lipolysis, enhancing sensory properties [41].

Counts of mesophilic bacteria, fungi, and yeasts in Queso Crema de Chiapas at three ripening stages (T1 = 2, T2 = 29, and T3 = 58 d). Bars with the same letter indicate no significant difference.

In this study, a direct relationship between culture-dependent and metagenomic methods was not established; however, total coliforms were found to be present at low abundance and nonviable. Mesophilic microorganisms were abundant throughout ripening, whereas fungal and yeast DNA was still detected at T3, even though these organisms appeared nonviable, as no growth was observed on culture media.

Principal Component Analysis (PCA) was performed to explore relationships between isolates and ripening stages. The resulting biplot (Fig 2) shows that PC1 accounts for 42.84% and PC2 accounts for 31.46% of the total variance. Together, they explain 74.30% of the variance, indicating a robust model that requires few explanatory variables. The ripening times are distributed across different quadrants; however, T1 and T2 are positioned closely together, suggesting that they may share similar characteristics.

Figs 3–5 show that T1 can be differentiated by its protein content, while T2 is distinguished by moisture and fat values. Protein content is highest at T1, with a value of 27.81%, and contributes positively to the separation of samples in the PCA biplot.

The variables humidity and fat did not show any statistical differences, but they contributed more positively to T2. This behavior can be explained by the nature of ANOVA, which compares variability between groups with variability within groups [42]. In contrast, principal components are used to reduce the dimensionality of a dataset, with the greatest variance typically captured by PC1 [43]. Finally, the samples belonging to T3 exhibit different physicochemical characteristics compared with those from T1 and T2.

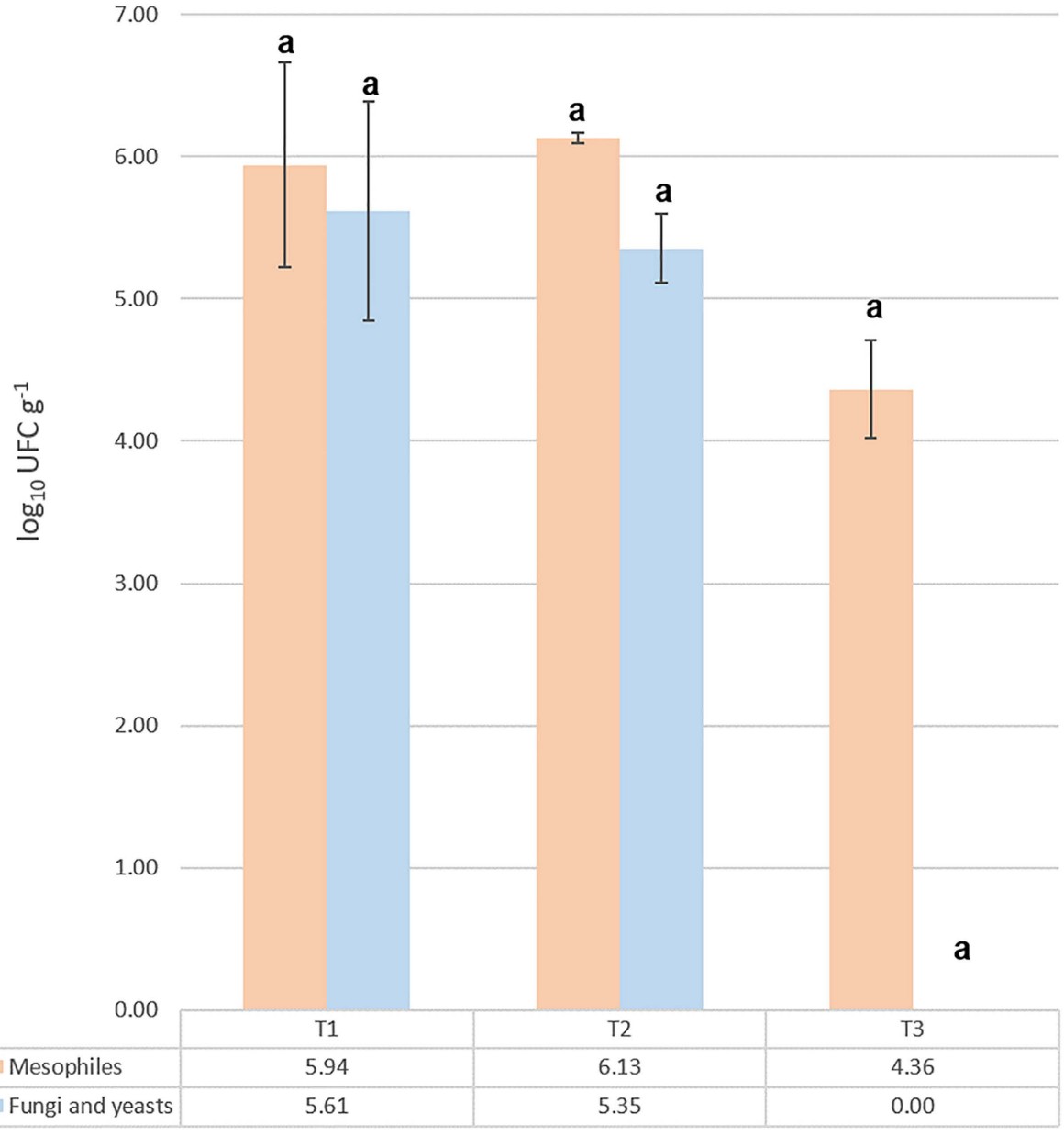

| | T1 | T2 | T3 |
|---|---|---|---|
| ■ Mesophiles | 5.94 | 6.13 | 4.36 |
| ■ Fungi and yeasts | 5.61 | 5.35 | 0.00 |

**Fig 1. Counts of mesophilic bacteria, fungi, and yeasts.**

PCoA (Fig 6), followed by PERMANOVA based on the Bray–Curtis dissimilarity matrix, confirmed the presence of statistically significant differences (p = 0.003) in the physicochemical and microbiological composition of Queso Crema de Chiapas across the three ripening times.

## Metagenomics

A total of seven bacterial genera were identified throughout the ripening process. At T1, the detected genera were *Streptococcus*, *Lactobacillus*, *Aeromonas*, *Lactococcus*, and *Enterobacter*; at T2, *Streptococcus*, *Lactobacillus*, *Lactococcus*,

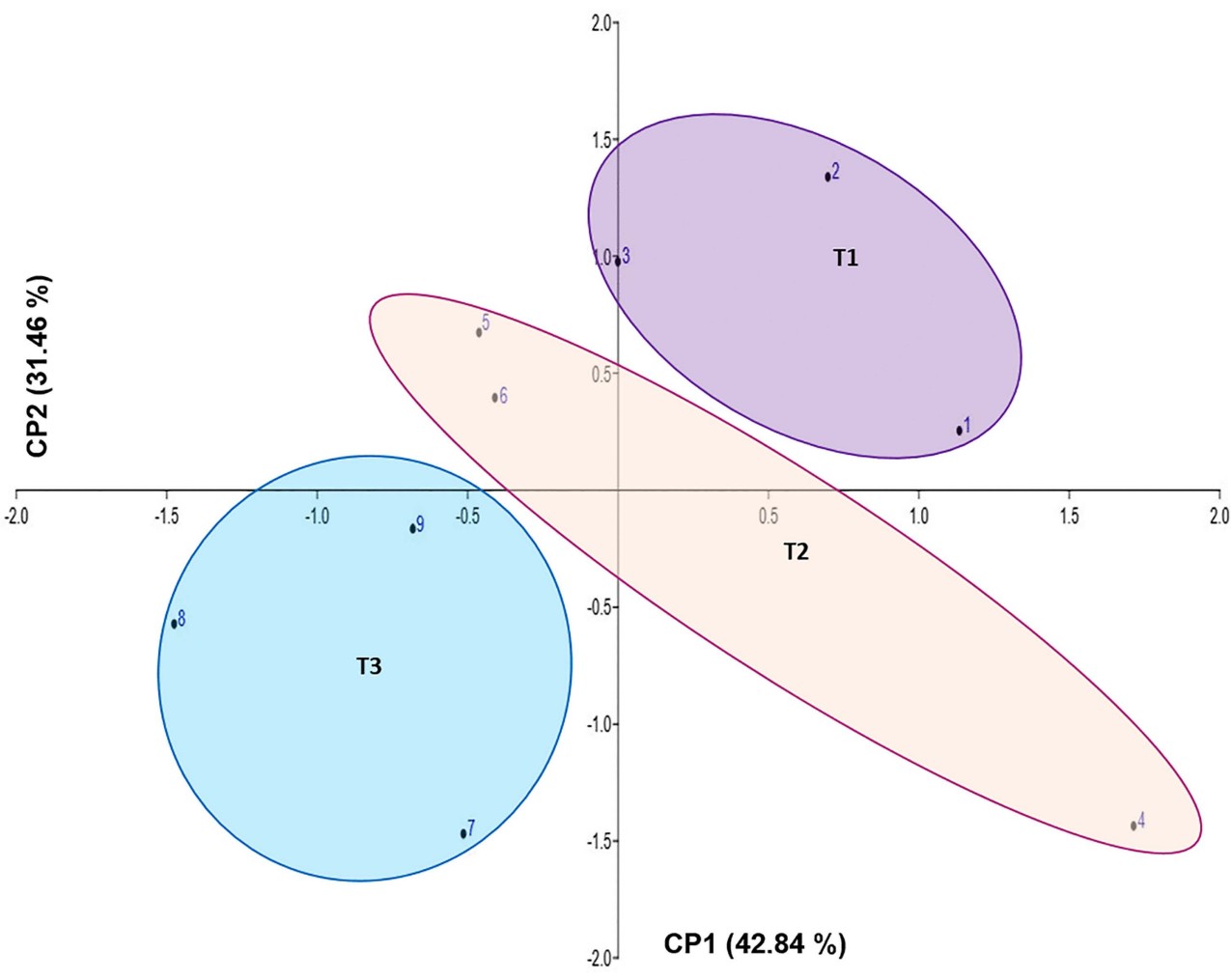

**Fig 2. PCA score plot grouped with K-means.**

*Aeromonas*, and *Chryseobacterium*; and at T3, *Streptococcus*, *Lactobacillus*, *Lactococcus*, *Aeromonas*, *Bifidobacterium*, and *Chryseobacterium* (Fig 7). Variations in microbial abundance during ripening are influenced by physicochemical and environmental factors that drive the selection of specific bacterial taxa [44]. In the present study, the number of genera increased slightly at 58 days (from five to six), a pattern consistent with observations in Dutch-type cheeses [45].

Relative abundance of bacteria in Queso Crema de Chiapas during ripening (T1 = 2 d, T2 = 29 d, and T3 = 58 d) based on the V3–V4 regions of the 16S gene.

Alpha diversity reflects the variety or richness of species within a community [46]. Shannon and Simpson indices are the most commonly used indicators [47]. These metrics, along with Chao1, are calculated based on species and ASV abundance [48]. The cheese at 2 d of processing showed the highest species richness and diversity (Table 4), as indicated by a Shannon value of 3.13—within the typical range of 1.5 to 4.5 [49]—and a Simpson index of 0.90, which falls between 0 and 1, suggesting high microbial diversity [50]. Similarly, a decrease in diversity was observed as ripening progressed; this behavior is consistent with findings in Parmigiano Reggiano cheese, where only microorganisms capable of using alternative energy sources to milk carbohydrates survive under low $a_w$ conditions [51]. Based on the Chao1 index,

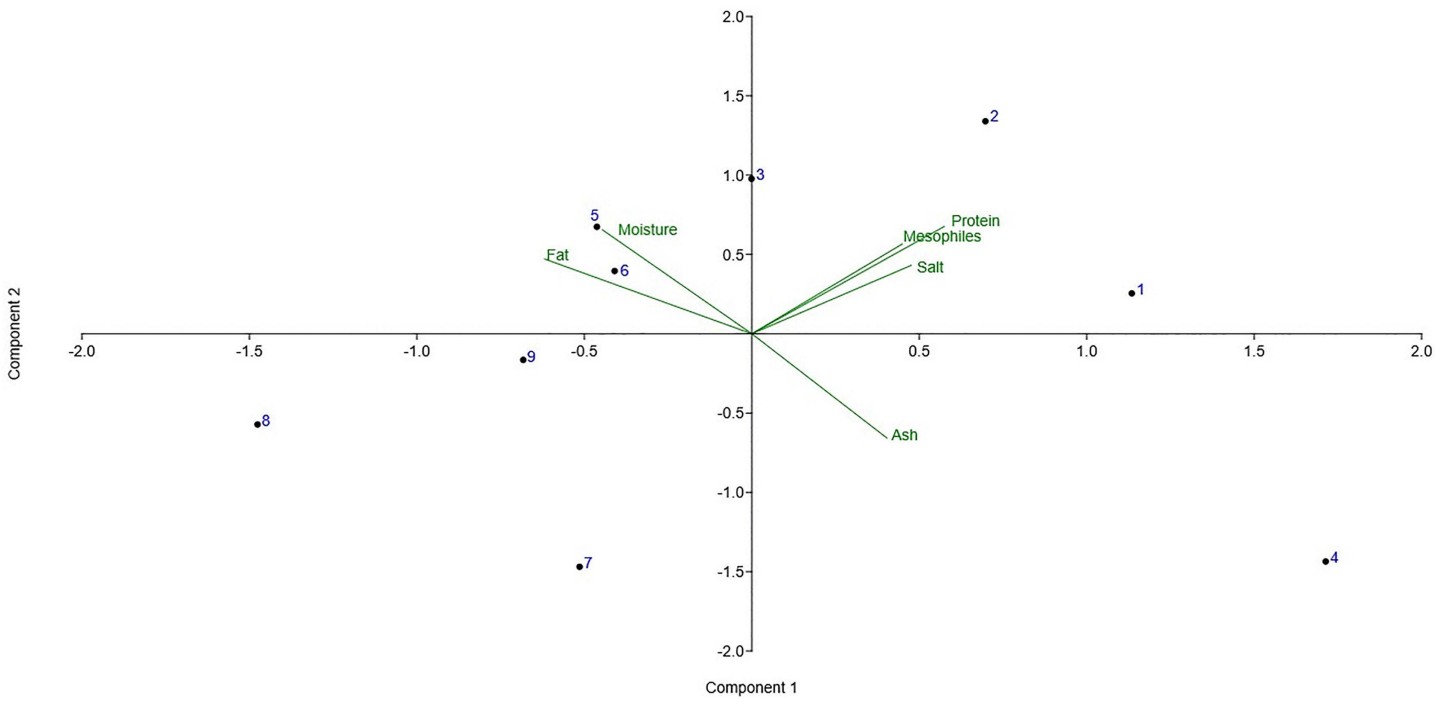

**Fig 3. PCA biplot.**

a decrease in species richness was also observed during ripening in Queso Crema de Chiapas, with the highest value (158) recorded at 2 days. However, this pattern contrasts with the findings of Barzideh *et al*. [50], where Chao1 increased with ripening time. This difference may be attributed to the physicochemical and microbiological characteristics of Queso Crema de Chiapas, which inhibited the growth of certain microorganisms while favoring others.

The most abundant genus in Queso Crema de Chiapas across the different ripening stages was *Streptococcus*, followed by *Lactobacillus* and then *Lactococcus*. This may be due to the presence of low–molecular weight nitrogenous compounds and its association with the genus *Lactobacillus* [52]. The genera *Streptococcus* and *Lactobacillus* have been reported in mexican cheeses such as Poro cheese and Bola de Ocosingo [52], as they are naturally present in milk and whey.

In Queso Crema de Chiapas, the second most abundant genus was *Lactobacillus*, which decreased over time. This behavior is consistent with what has been reported in Tulum cheese [53] and Feta cheese [54], where high salt content (7%) and low pH (4.4) affected the survival of this genus. Changes in *Lactobacillus* abundance during ripening are associated with modifications in physicochemical characteristics such as total solids, protein, and pH, as previously described in Greek Feta cheese [33] and Parmigiano Reggiano [51].

*Lactococcus* was the third most abundant genus; at 58 d it had a relative abundance (RA) of 0.25. This trend is similar to findings in Gouda cheese [55] and Feta cheese [56], which showed abundances of 58.5% and 90%, respectively. *Lactococcus* is not highly demanding with respect to temperature, as it can grow between 10–40 °C and tolerates high salt concentrations [57]. The genera *Streptococcus*, *Lactobacillus*, and *Lactococcus* can contribute to the sensory properties of Queso Crema de Chiapas throughout ripening. In Cantal and Historic Rebel cheeses, *Lactobacillus* subsp. was correlated with esters and alcohols such as 2-heptanol, derived from ketone reduction, while *Lactococcus* subsp. was correlated with ketones [19], [58]. Esters are associated with fruity, sweet, floral, and spicy notes [58], and n-alcohols are also associated with fruity notes, often resulting from fatty acid lipolysis [59], [58].

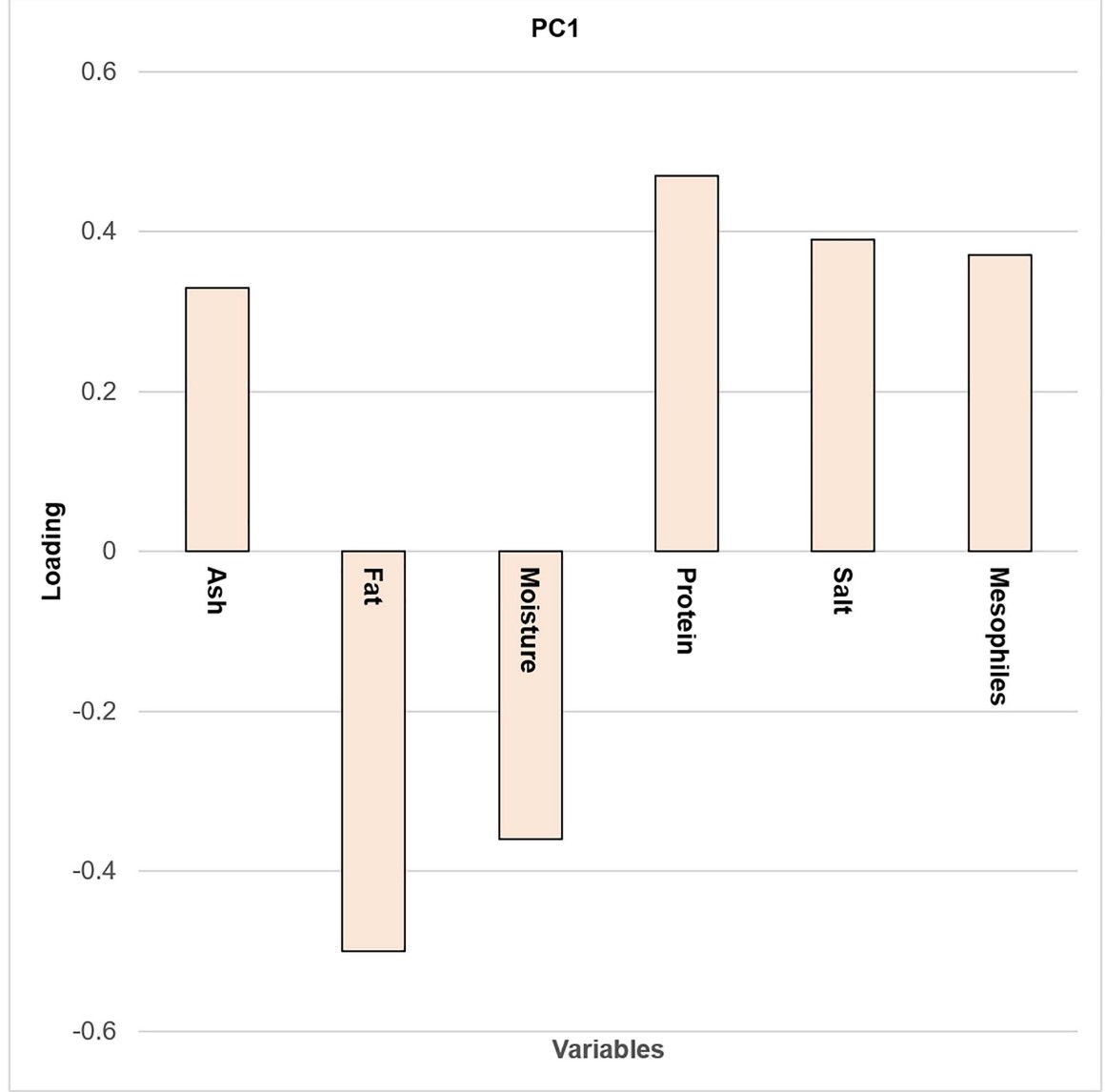

**Fig 4. Bar chart of PCA loadings for PC1.**

In Queso Crema de Chiapas, *Aeromonas*, *Enterobacter*, *Chryseobacterium*, and *Bifidobacterium* were also found in lower RA. *Aeromonas* may enter the product due to poor production practices, and its abundance decreases during ripening, consistent with what has been reported for Paipa cheese [60]. *Aeromonas* species have been identified in Panela cheese, Adobera cheese, Ranchero cheese, Jocoque [61], and fresco cheese from Sonora [62]. Some *Aeromonas* strains possess virulence genes that, when combined with virulence factors from other strains, can cause severe diarrhea in humans [63]. *Enterobacter* was detected at T1 and decreased throughout ripening, similar to what has been observed in soft and semi-hard cheeses [64], mainly due to physicochemical changes occurring during the ripening process [65].

At T2 and T3, *Chryseobacterium* was detected, a genus previously reported in dairy products and raw milk by Hugo et al. [66], as well as in Poro cheese from Tabasco [44]. Members of this genus are emerging opportunistic pathogens and,

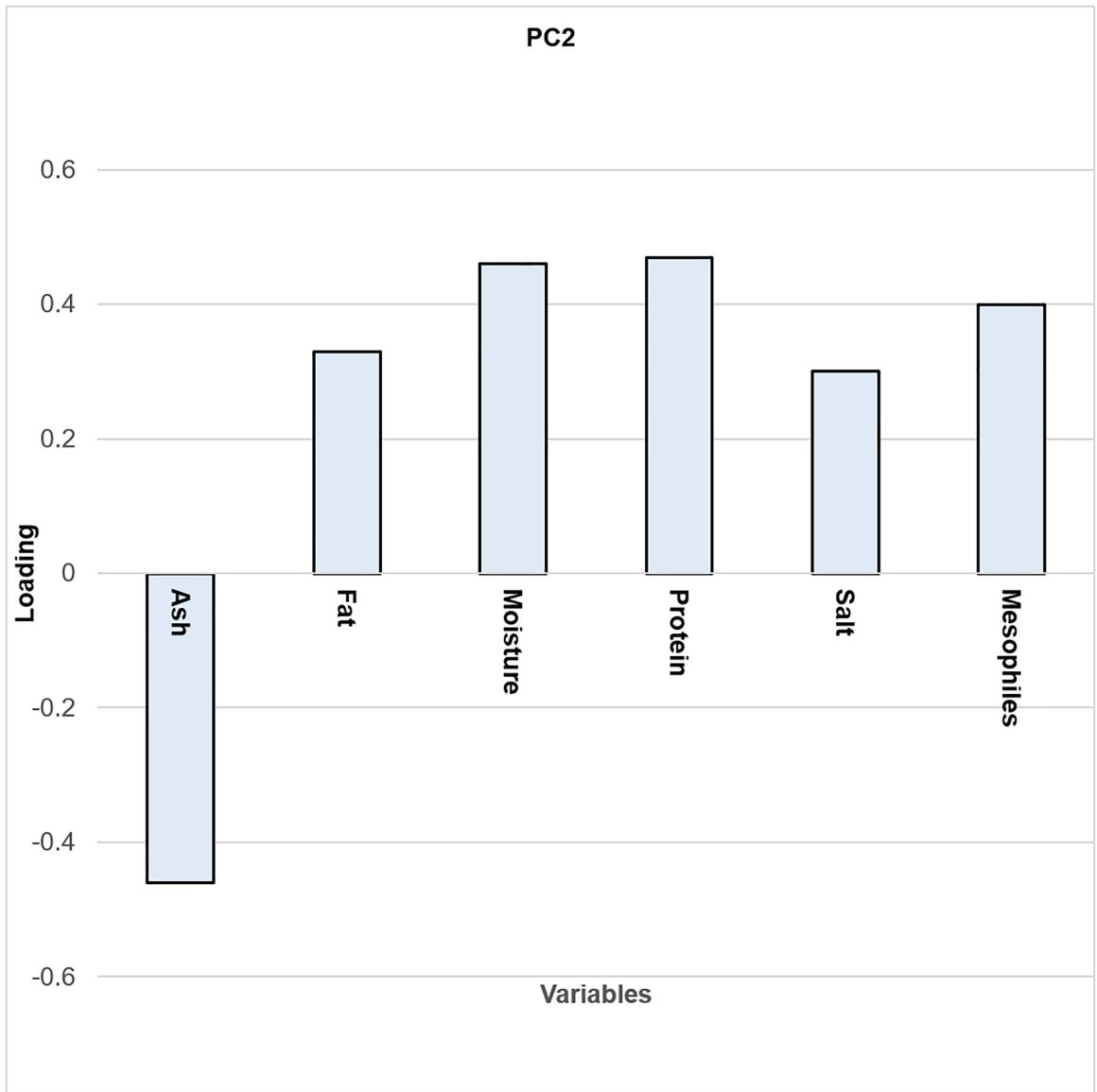

**Fig 5. Bar chart of PCA loadings for PC2.**

together with other bacterial groups, can serve as indicators of production environment conditions [67]. In milk, they have been associated with the production of yellow pigments [68], while in Fontina cheese they contribute to sensory characteristics through proteolytic and lipolytic activities [69].

Another important genus found at the third ripening stage was *Bifidobacterium*, which has also been reported in Italian cheeses [70], Parmigiano Reggiano [51], and "Tomme d'Orchies" [71]. This genus is relevant due to its association with 11 documented bioactivities [72]. However, to exert such benefits, it must be present in sufficient quantities, exhibit adhesion capability, and survive intestinal conditions [72].

To illustrate the distribution of the 213 amplicon sequence variants (ASVs) identified across ripening stages, a Venn diagram was generated (Fig 8). The data showed that 102 ASVs—representing 48% of total reads—were shared among all

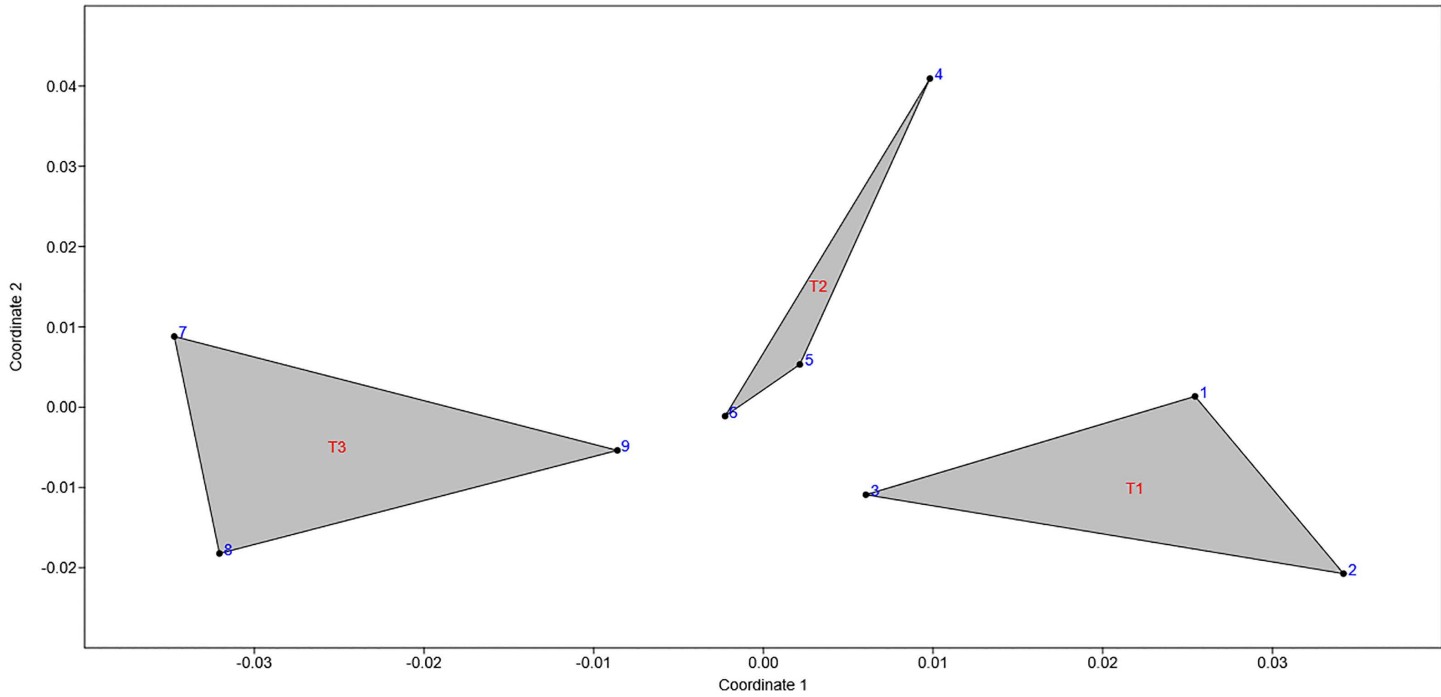

**Fig 6. PCoA plot using Bray–Curtis distance.**

three ripening stages. Thirty-six of the ASVs accounted for 17% and represent microorganisms shared between two ripening stages (T1–T2, T2–T3, T3–T1), while 75 ASVs, corresponding to 35%, are unique ASVs. The highest number of unique ASVs (n = 28) was observed at the 58 d ripening stage (T3), while the lowest number (n = 22) was found at 2 d (T1). This behavior may be explained by the lower NaCl content at T3, which favored the presence of bacteria capable of affecting product quality by generating unpalatable compounds [73]. Shared ASVs with higher abundance can provide information about bacteria that could be selected as potential starter cultures [74], as well as pathogenic bacteria that persist throughout ripening. In contrast, time-specific bacteria contribute to the unique characteristics of the cheese at each ripening stage.

Venn diagram showing the number of specific and shared ASVs among the three ripening stages of Queso Crema de Chiapas.

## Fungal and yeast dynamics in Queso Crema de Chiapas

As indicated in Table 5, the Shannon, Simpson, and Chao1 indices revealed that fungal and yeast richness and diversity increased with ripening time, reaching their maximum values at 58 days (Shannon = 1.81, Simpson = 0.64, and Chao1 = 48.88). In the study by Dimov et al. [75], it was reported that in Bulgarian green cheese, the Shannon and Chao1 index values similarly increased with ripening time. Levante *et al*. [76] also found that microbial diversity in buffalo Mozzarella cheese varied depending on the place of production.

The most common genera found in different cheeses are *Candida*, *Pichia*, *Saccharomyces*, and *Trichosporon* [77]. Sixteen fungal genera have been identified in Mexican cheeses, with the most abundant being *Galactomyces*, *Saccharomyces*, and *Scheffersomyces* [61].

In Queso Crema de Chiapas, eight fungal species and two yeast species were identified throughout the ripening process. At T1, the following genera were detected: *Saccharomyces cerevisiae* (0.47 RA), *Candida versatilis* (0.5 RA), *Candida tropicalis* (0.016 RA), and *Candida etchellsii* (0.014 RA). At T2, the most abundant species was *C. versatilis* (0.88

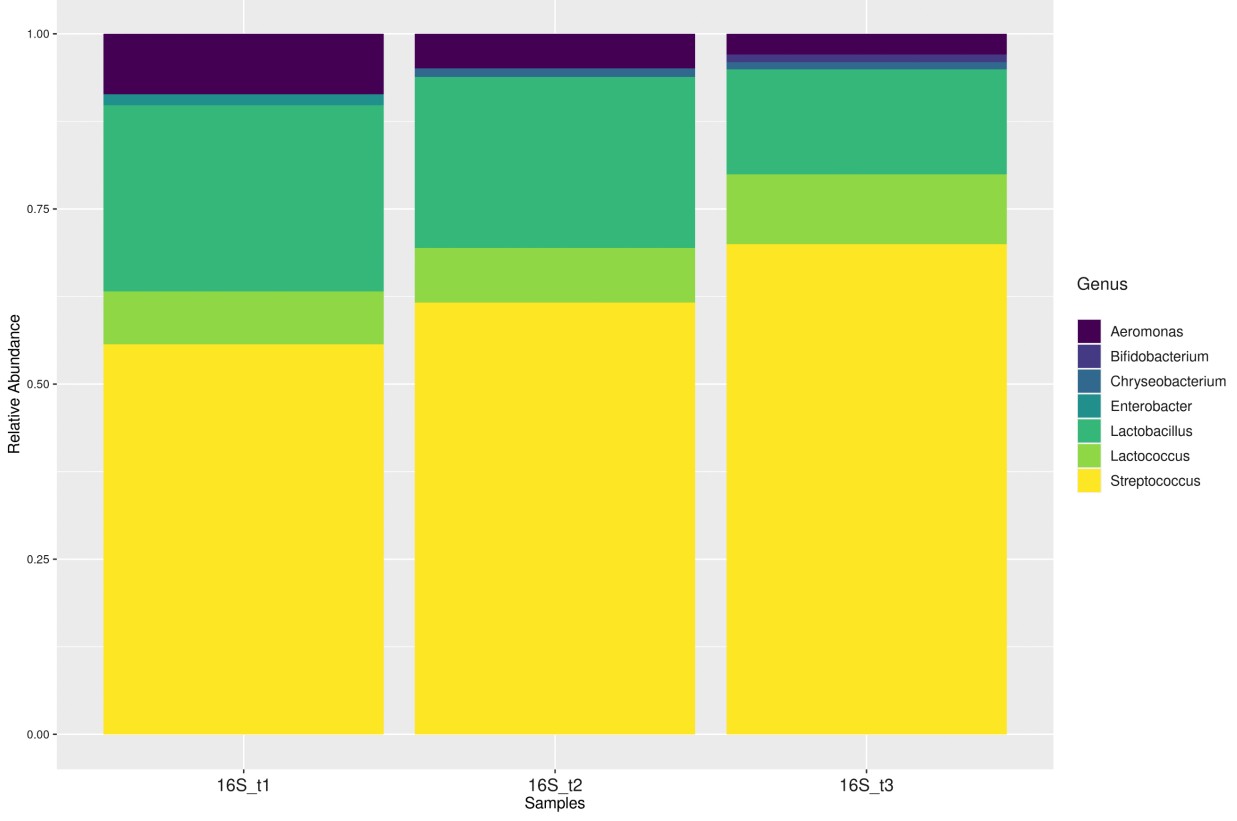

**Fig 7. Relative abundance of bacteria in Queso Crema de Chiapas.**

**Table 4. Alpha diversity for 16S.**

| Sample | Observations | Chao1 | Shannon | Simpson |
|---|---|---|---|---|
| T1 | 133 | 158 | 3.13 | 0.90 |
| T2 | 136 | 147.55 | 3.08 | 0.89 |
| T3 | 132 | 143 | 2.90 | 0.88 |

Chao1, Shannon, and Simpson indices at three ripening stages (T1 = 2 d, T2 = 29 d, and T3 = 58 d) based on the 16S gene in Queso Crema de Chiapas.

RA); in addition, *Candida kruissii* (0.062 RA), *C. etchellsii* (0.03 RA), *Phaeoacremonium hungaricum* (0.013 RA), and *unclassified Pichia* (0.025 RA) were observed. At T3, *C. versatilis* remained the most abundant species (0.625 RA); other species identified included *C. kruisii* (0.0625 RA), *C. etchellsii* (0.125 RA), *C. tropicalis* (0.042 RA), *P. hungaricum* (0.042 RA), *Pichia (unclassified)* (0.042 RA), *Trichosporon debenedettiannum* (0.022 RA), *Malassezia furfur* (0.022 RA), *S. cerevisiae* (0.011 AR), and *Candida parapsilosis* (0.011 RA).

A greater number of fungal species were observed as ripening progressed. This trend is similar to what has been reported for Cantal cheese [19] and Austrian Vorarlberger Bergkäse cheese [48]. Differences in species abundance were attributed to the source of the cream used, the presence of polyunsaturated fatty acids, n-alcohols, and production facilities [48], [19].

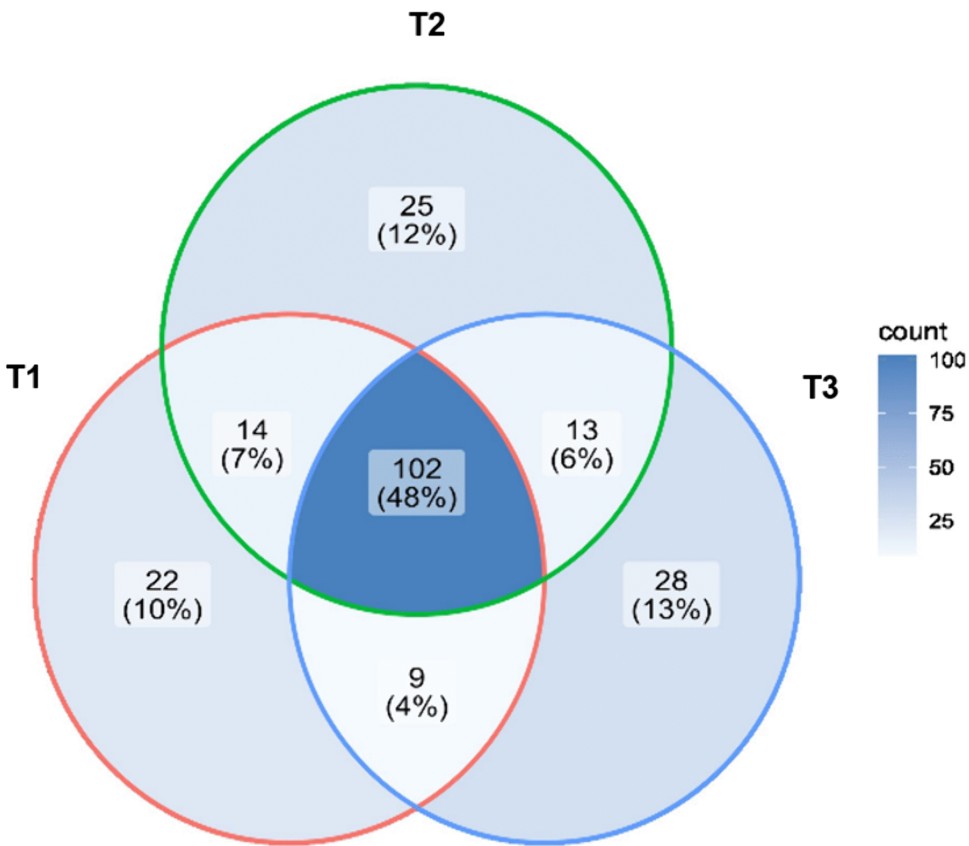

**Fig 8. Venn diagram.**

**Table 5. Alpha diversity for ITS.**

| Sample | Observations | Chao1 | Shannon | Simpson |
|---|---|---|---|---|
| **T1 ITS** | 33 | 35.8 | 1.27 | 0.60 |
| **T2 ITS** | 38 | 41 | 1.06 | 0.36 |
| **T3 ITS** | 42 | 48.88 | 1.81 | 0.64 |

Chao1, Shannon, and Simpson indices at three ripening times (T1 = 2 d, T2 = 29 d, and T3 = 58 d) determined using ITS region sequencing.

Across all three ripening stages of Queso Crema de Chiapas, the three most abundant fungal strains were *C. versatilis*, *C. etchellsii*, and *C. tropicalis*. Among these, *C. versatilis* was not only the most abundant but also the most variable over time. In Vorarlberger Bergkäse cheese, the genus *Candida* was abundant at the beginning of ripening and decreased over time [48]. *C. versatilis* has been identified in whey, yogurt, and cheeses [78], including Gorgonzola and Danish cheeses [79]. It is a salt-resistant strain; although its role in cheese is not fully understood, it is known to produce glycerol and mannitol when metabolizing glucose [80]. *C. etchellsii* was the second most abundant strain, and its abundance decreased as ripening progressed. This strain has been reported in Cotija cheese [81] and Taleggio cheese after seven days of ripening [18]. Although its specific contribution in cheese is not fully understood, it has been shown to participate in ester synthesis in Chinese bean chili paste, contributing to flavor development [82].

*C. tropicalis* was the third most abundant strain and its abundance increased with ripening time. This strain has been identified in Oaxaca cheese [83], raw bovine milk, Cremoso cheese [84], and Mihalic cheese [85]. Dinika *et al*. [86] reported that during the fermentation of Mozzarella cheese whey with *C. tropicalis*, peptides were hydrolyzed, generating amino acids such as Asp, Glu, Thr, Val, Ile, and Lys, which are associated with glutamate biosynthesis as an amine donor.

*S. cerevisiae* showed the highest RA value (0.47) in Queso Crema de Chiapas at T1 (Fig 8). This yeast has been identified in Frescal cheese [87], as well as in Gorgonzola and Danbo cheeses, where it contributes to the production of aldehydes and branched-chain alcohols [88]. *Saccharomyces* has also been reported by Murugesan *et al*. [61] in various Mexican cheeses, including Adobera, Doble Crema de Chiapas, Ranchero, Chihuahua, Cincho, Oaxaca, Canasta, Manchego, and Jocoque.

Other strains were found in lower abundance in the Queso Crema de Chiapas samples, including *C. kruisii*, *P. hungaricum*, *unclassified Pichia*, *T. beermannianum*, *M. furfur*, and *C. parapsilosis*. The abundance of *C. kruisii* decreased as ripening progressed. *P. hungaricum* was detected at T3; although this genus has not been reported in cheeses, it has been isolated from the environment, diseased woody plants, wood, grapevines, humans with phaeohyphomycotic infections, bark beetle larvae, arthropods, and soil [89].

The genus *Pichia* was detected at the third ripening stage with an RA of 0.06 (Fig 9). This behavior is opposite to what has been reported in Fossa cheese, where *Pichia occidentalis* was found in the production environment and decreased during ripening [90]. The *Pichia* genus has been identified in Mexican artisanal cheeses [61], in Cotija cheese [81], and in Roquefort cheese [91]. It has also been used as a starter culture in Cantal cheese [92]. In Kasajo cheese, it was observed that different strains produce different aroma profiles; for example, the presence of *Pichia kudriavzevii* A11 was associated with alcohols, acetic acid, and acetates [93].

In Queso Crema de Chiapas, *T. debeurmannianum* was detected at the T3 ripening stage. This genus has been reported in raw milk and Quebec cheese [94]. Its presence indicates contamination, as *T. debeurmannianum* has been identified in clinical samples from patients with urinary tract infections [95], diabetic foot infections [96], fungal ampoules [97], as well as in the environment, skin, intestinal tract, and vagina [98]. The genus *Trichosporon* utilizes various carbohydrates and carbon sources and degrades urea, but it is not considered a fermentative microorganism [98].

In the cheese analyzed, *M. furfur* was found at T3 (Fig 9). This species is uncommon in cheese; however, the genus *Malassezia* is lipid-dependent [99]. Because it can grow under aerobic or anaerobic conditions at temperatures ranging from 33 to 41°C [100], it is commonly found on human skin [101] and in breast milk [102]. Its presence in dairy products has not been sufficiently studied to determine whether it can be considered an indicator of contamination.

*C. parapsilosis* was identified at low abundance (0.031) at T3. This species has been shown to promote the growth of *Lactobacillus paracasei* in Comté cheese [103] and has been detected in Cheddar cheese, Swiss-type cheese, and American blue cheese [104]. Its presence contributes to the hydrolysis of α-lactalbumin and β-lactoglobulin, and it mainly produces alcohols [103].

Relative abundance of fungi and yeasts in Queso Crema de Chiapas during ripening (T1 = 2 d, T2 = 29 d, and T3 = 58 d), detected through ITS1 and ITS2 region sequencing.

To evaluate the distribution of the 72 fungal and yeast ASVs identified across the ripening stages, a Venn diagram was generated (Fig 10). The data showed that 30 ASVs—representing 42% of the total reads—were shared among all three ripening times. Ten ASVs (14%) were shared between two ripening times, while 32 ASVs (44%) appeared uniquely in each ripening stage. The highest number of unique ASVs (n = 13) was found at T1 and T3, whereas the lowest number (n = 6) was observed at T2. The number of fungal species at T1 and T3 differed by only one species, which may be explained by the osmotolerance of some fungi; for example, *S. cerevisiae* has been reported to produce glycerol to tolerate stress conditions [105]. On the other hand, the lower number of species observed at T2 could be related to the lack of changes in physicochemical characteristics at this stage. One possible explanation is that bacteria present at T2 may have inhibited certain fungal strains. Leyva-Salas *et al*. [106] reported that organic acids produced by *Lactobacillus* strains inoculated into semi-hard cheese and yogurt exhibited antifungal activity against *Penicillium*, *Candida*, and *Trichosporon*.

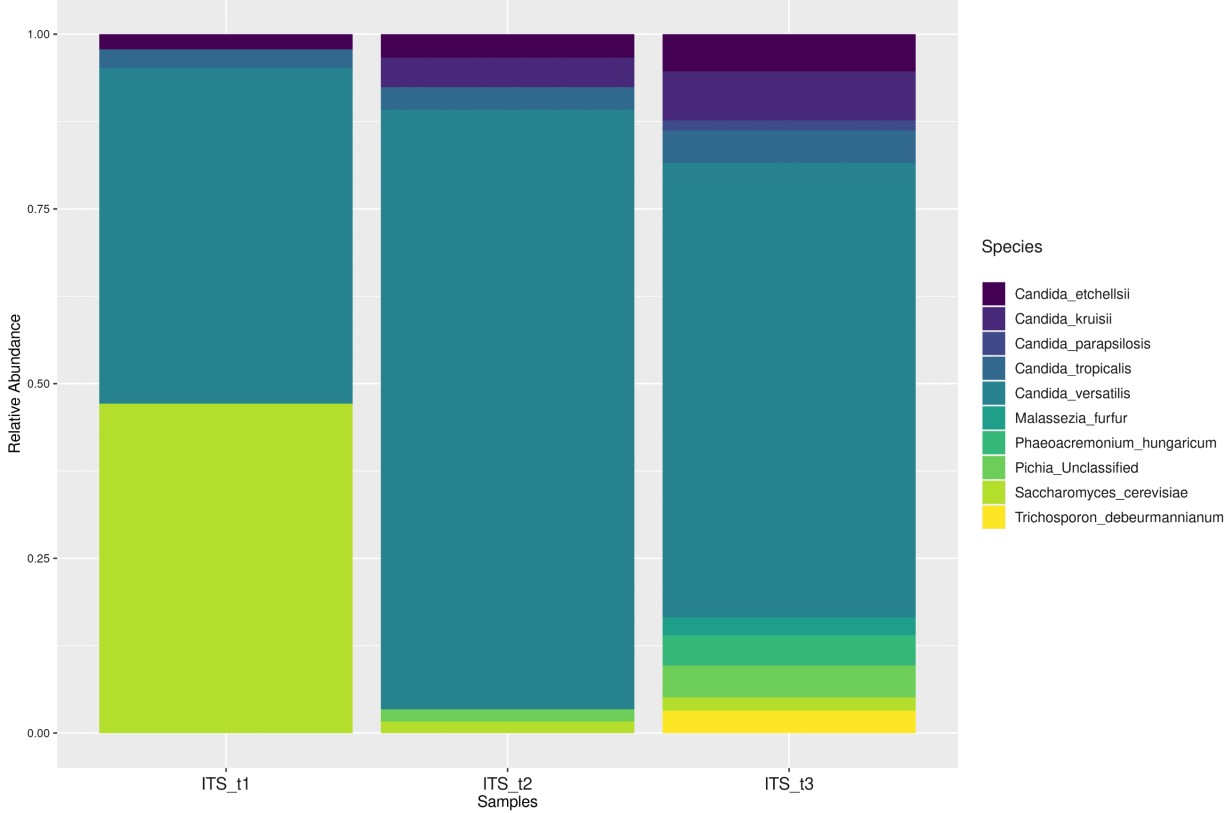

**Fig 9. Relative abundance of fungi and yeasts.**

Venn diagram showing the number of specific and shared ASVs among the three ripening stages (2, 29, and 58 days) of Queso Crema de Chiapas made from raw cow's milk.

### Integrated analysis reveals a clear ecological succession driven by ripening time

To comprehensively characterize microbial community dynamics, an integrated heatmap was constructed using bacterial (16S rRNA) and fungal (ITS) sequencing data at the genus level (Fig 11). This approach allows for the visualization of ecological patterns and interactions that shape the microbiota during cheese ripening.

A key outcome of this analysis was the clear evidence of ecological succession throughout ripening. The hierarchical clustering dendrogram (top) distinctly separates the early ripening stage (T1, 2 days) from the later stages (T2, 29 days; T3, 58 days), indicating a substantial shift in microbial community structure over time. The close clustering of T2 and T3 suggests that the microbiota transitions toward a more stable and mature composition following the initial colonization phase.

Furthermore, the clustering of taxa shown in the left dendrogram reveals co-occurrence patterns by grouping microorganisms with similar abundance profiles across sampling times. This observation suggests the presence of functional guilds or ecological dependencies among specific bacterial and fungal genera. Temporal changes in microbial community composition correlate directly with ripening progression and associated shifts in protein content and alpha diversity (Shannon index), reinforcing the conclusion that maturation is a primary driver of microbial selection.

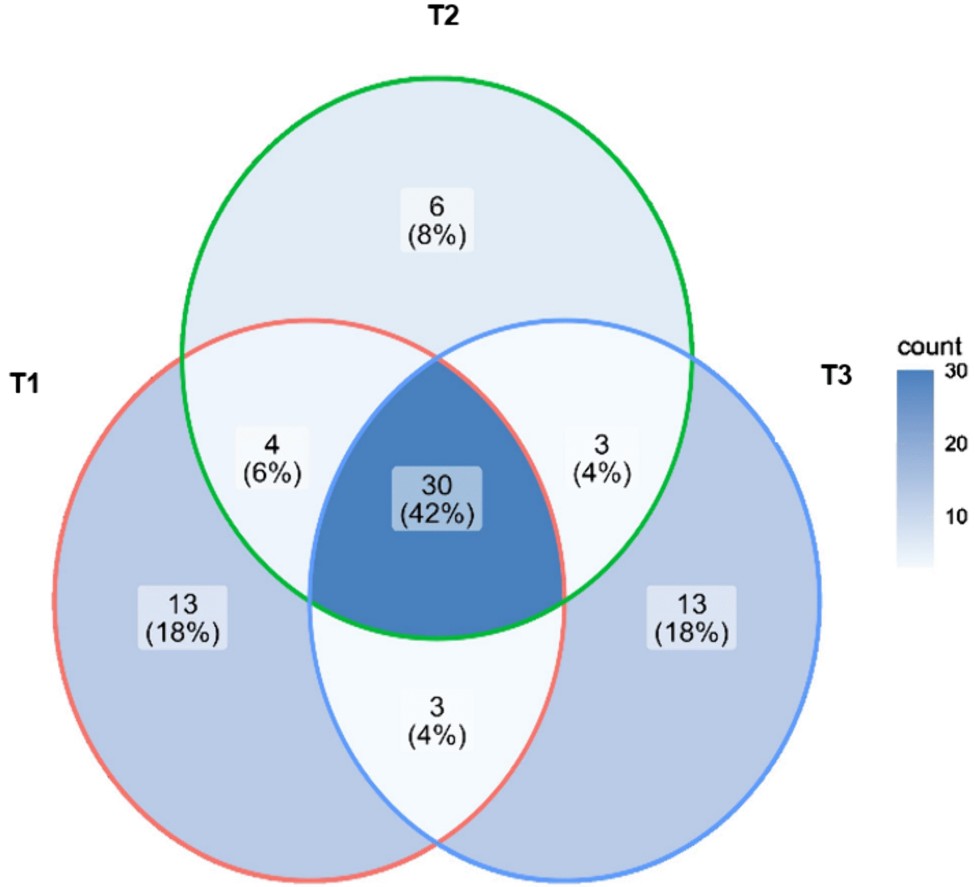

**Fig 10. Venn diagram showing the number of specific and shared ASVs.**

Each row represents an individual Amplicon Sequence Variant (ASV), and each column represents a sample (T1 = 2 d, T2 = 29 d, T3 = 58 d). Colors indicate the row-normalized relative abundance (Z-score), where red signifies higher abundance and blue signifies lower abundance compared to the means of that ASV across all samples. The top dendrogram clusters samples based on their overall microbial profile similarity (Bray–Curtis dissimilarity). The left dendrogram clusters ASVs with co-varying abundance patterns. The top annotation bars display metadata for each sample: ripening time, Shannon diversity index, and protein content. The left annotation bar (Genus) indicates the taxonomic assignment for each ASV.

Some observations from this study should be considered: (i) potential errors in species-level classification due to methodological constraints, as only the 16S rRNA and ITS regions were sequenced; greater taxonomic resolution could have been achieved through whole-genome sequencing; (ii) possible sequencing errors [107]; (iii) ongoing updates to reference databases [108]; and (iv) the methodological decision to pool DNA from biological replicates to generate a single representative consensus profile per experimental condition. In particular, the lack of within-group replicates markedly reduces the statistical power necessary for robust beta diversity assessments. Consequently, to maintain the integrity of our conclusions, diversity analyses were restricted to alpha diversity metrics, which still provide valuable insights into the richness and evenness within each representative pooled sample.

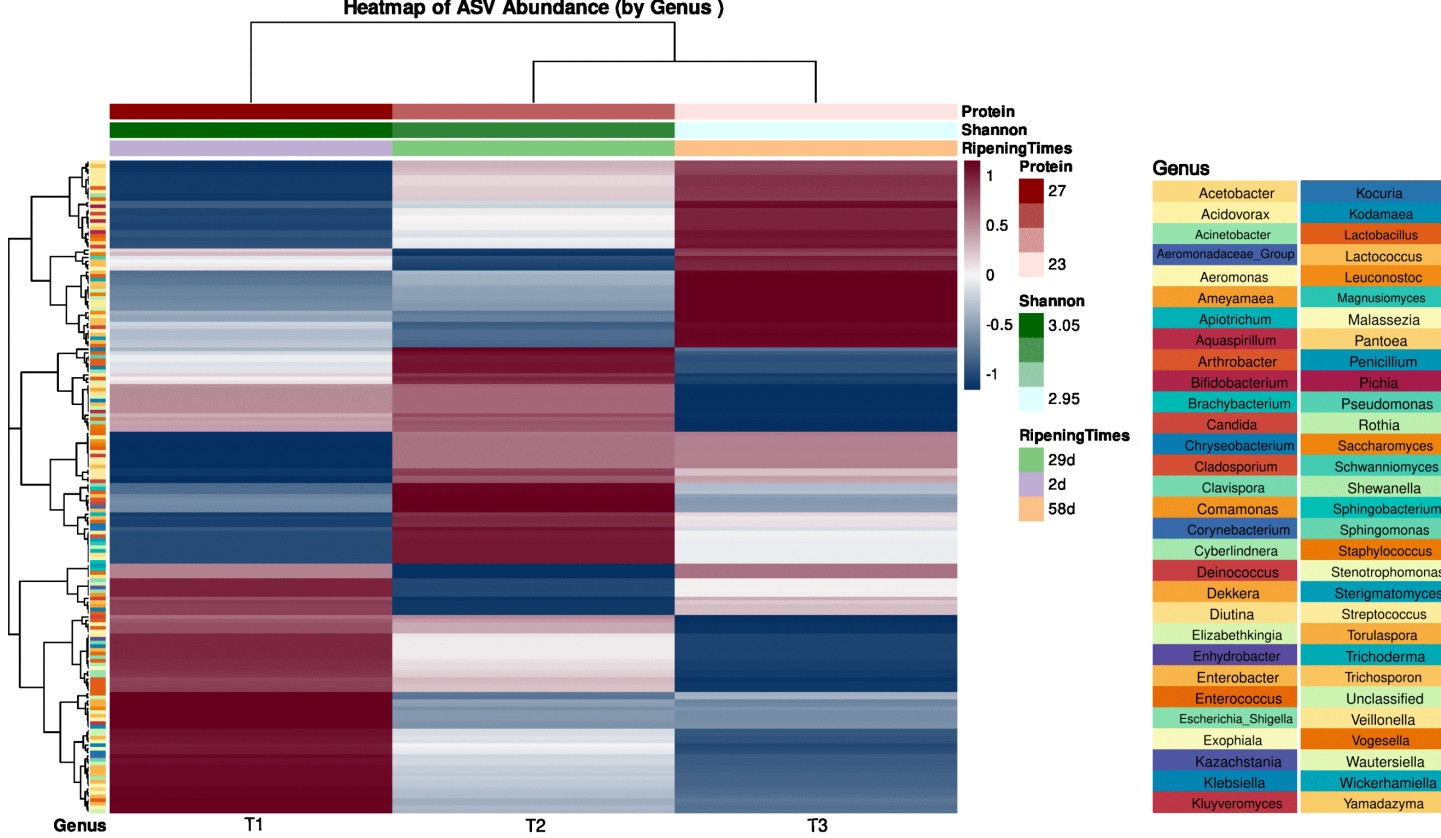

**Fig 11. Heatmap of ASV relative abundance showing microbial community patterns during ripening.**

Fig 12 displays a representative qPCR amplification curve, comprising three characteristic phases: (i) an exponential phase marked by low initial product concentration and abundant reaction components, (ii) a linear phase during which product accumulation increases and reaction efficiency gradually declines, and (iii) a plateau phase where amplification ceases once reagents are exhausted [109], [110]. In qPCR, the presence of a microbial genus or species was inferred from the cycle threshold (Ct) value and the shape of the amplification curve. According to Akingbola et al. [111], Ct represents the number of cycles required for fluorescence to reach the threshold baseline; lower Ct values indicate higher initial DNA concentrations, whereas higher Ct values reflect lower abundance.

A Ct value < 20 was interpreted as a positive detection of the target microorganism's DNA, reflecting a sufficiently high initial concentration for amplification. Conversely, Ct > 20 was considered negative, indicating DNA levels below the detectable threshold. When no rise above baseline fluorescence was observed, amplification was deemed absent, confirming the microorganism's non-detection.

Fig 12 illustrates amplification curves corresponding to one bacterial genus and two yeast species. *Chryseobacterium* was detected at the T2 ripening stage with a Ct value of 3.26; *Candida etchellsii* at T1 with a Ct of 7.22; and *Candida tropicalis* at T3 with a Ct of 9.84. The negative control (water) showed a Ct of 33.47, suggesting a late signal likely attributable to primer-dimer formation, which is not biologically meaningful. Ct values for the internal positive control were 21.87 at T1, 21.19 at T2, and 20.08 at T3. Ct values near 20 cycles indicate low gene concentration and are therefore not biologically significant.

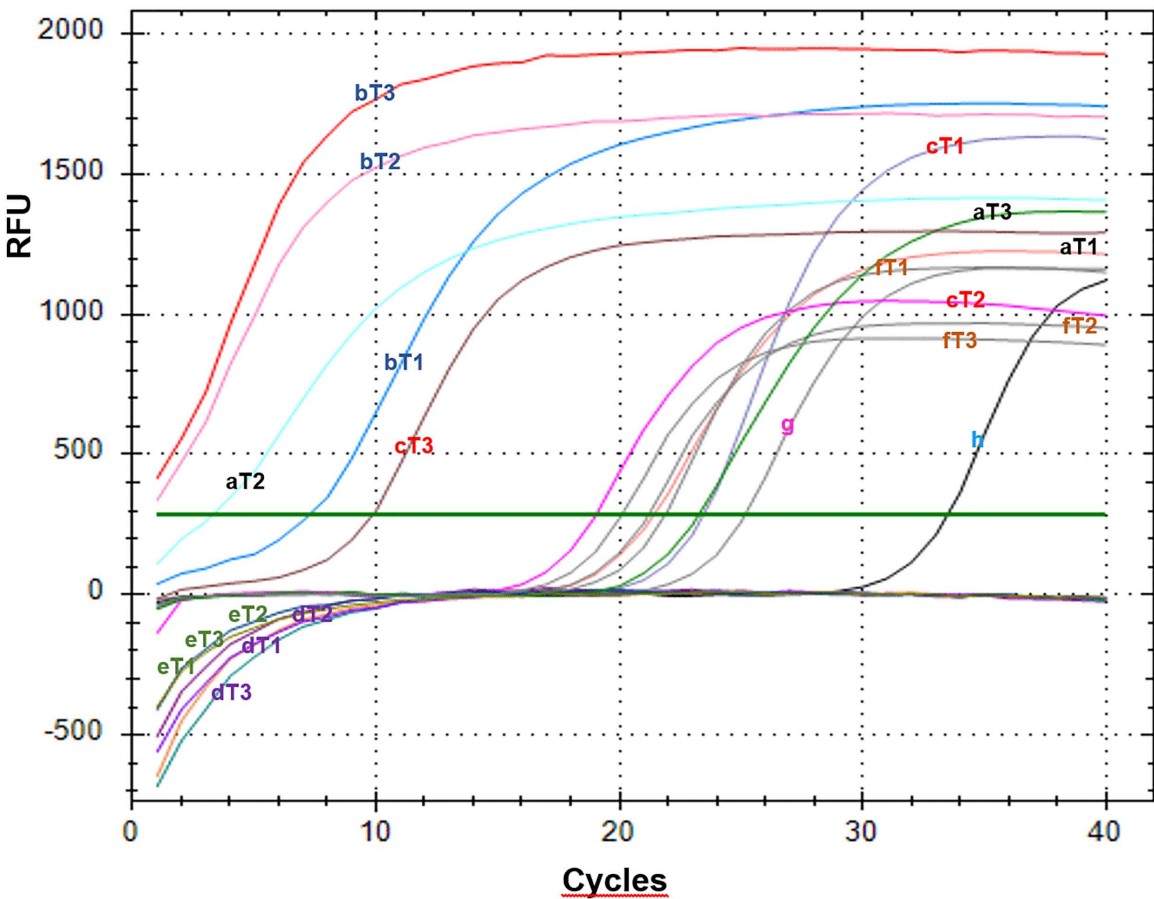

**Fig 12. qPCR amplification plots for three bacterial genera and two yeast species.**

The presence and absence of microorganisms varied throughout the ripening process. *Chryseobacterium* was detected at T2 (Fig 12), likely favored by the physicochemical characteristics and specific microbial interactions at that stage. However, its presence in foods is undesirable, as species within this genus are associated with spoilage due to their psychrophilic and proteolytic nature; some strains also produce biogenic amines and show antibiotic resistance [112]. In queso Crema de Chiapas, pathogenic bacteria such as *Aeromonas* and *Enterobacter* were not detected at any ripening stage (Fig 12), as their qPCR curves did not cross the baseline threshold, indicating absence of detectable DNA.

The presence of *Candida etchellsii* at T1 was likely related to the protein and salt content characteristic of this early ripening stage. It may later become inhibited by compounds produced by other microorganisms. *Candida etchellsii* has been identified as an osmotolerant strain due to glycerol accumulation and enhanced membrane transport capacity [113]. Its presence in queso Crema de Chiapas is desirable, as it contributes to characteristic sensory traits through lipolysis- and proteolysis-derived flavors and aromas.

*Candida tropicalis* was detected at T3, suggesting spoilage and potential contamination of the product. Therefore, we recommend avoiding prolonged storage of the cheese to reduce risks associated with the growth of undesirable microorganisms. *Candida tropicalis* has been associated with various human infections, including conditions affecting the skin,

oral cavity, ears, genital tract, and nails [114]. In foods, its presence is linked to spoilage and decomposition, making its detection in food matrices undesirable [115]. Environmental isolates of *C. tropicalis* have shown resistance to azole drugs; due to specific mutations, alternative pathways are formed that prevent the accumulation of toxic intermediates while maintaining cell membrane integrity [114], [116].

*Candida tropicalis* is an osmotolerant species due to mechanisms such as intracellular glycerol accumulation—acting as a compatible osmolyte—and regulation of membrane transporters such as $Na^+/K^+$-ATPase, which promote rapid ion efflux to restore intracellular osmotic balance [117]. However, even though these microorganisms were detected, it remains unknown whether they were viable or whether only their DNA was detected.

qPCR amplification plots across three ripening stages. a) *Chryseobacterium* (T1, T2, T3); b) *Candida etchellsii* (T1, T2, T3); c) *Candida tropicalis* (T1, T2, T3); d) *Enterobacter* (T1, T2, T3); e) *Aeromonas* (T1, T2, T3); f) Internal positive control GAPDH (T1, T2, T3); g) External positive control GAPDH; h) Water control.

Some of the microorganisms identified through 16S and ITS amplicon sequencing do not necessarily match the qPCR results in queso Crema de Chiapas (Table 6); that is, some taxa were detected by NGS but appeared at low relative abundance by qPCR [118]. The opposite can also occur: NGS may fail to detect microorganisms present at low abundance. This discrepancy can depend on several factors, such as read length, adapters, and the reference databases used, among others [119], [118]. In addition, Tettamanti Boshier et al. [119] reported that qPCR exhibits higher sensitivity than NGS for small amounts of bacterial DNA. It is therefore recommended to complement amplicon-based approaches with qPCR or plate culturing, as the former may overrepresent certain bacterial groups and fungal strains.

**Table 6. Presence of microorganisms in Queso Crema de Chiapas identified by qPCR and NGS.**

| Microorganism | Time | Cq | qPCR | Presence/Absence 16S and ITS |
|---|---|---|---|---|
| *Chryseobacterium* | T1 | 21.40 | – | – |
| *Chryseobacterium* | T2 | 3.26 | + | + |
| *Chryseobacterium* | T3 | 23.25 | – | + |
| *Candida etchellsii* | T1 | 7.22 | + | + |
| *Candida etchellsii* | T2 | | – | + |
| *Candida etchellsii* | T3 | | – | + |
| *Candida tropicalis* | T1 | 23.44 | – | + |
| *Candida tropicalis* | T2 | 19.03 | – | + |
| *Candida tropicalis* | T3 | 9.84 | + | + |
| *Enterobacter* | T1 | | – | + |
| *Enterobacter* | T2 | | – | – |
| *Enterobacter* | T3 | | – | – |
| *Aeromonas* | T1 | | – | + |
| *Aeromonas* | T2 | | – | + |
| *Aeromonas* | T3 | | – | + |
| Control positivo interno GAPDH | T1 | 21.87 | – | |
| Control positivo interno GAPDH | T2 | 21.19 | – | |
| Control positivo interno GAPDH | T3 | 20.08 | – | |

## Conclusions

This study characterizes the microbial dynamics of the Mexican artisanal cheese "Queso Crema de Chiapas" using metagenomic analysis. Microbial and fungal communities were examined at three distinct ripening stages (2, 29, and 58 days), revealing temporal shifts in community composition. The most abundant taxa identified through amplicon sequencing included *Streptococcus*, *Lactobacillus*, *Lactococcus*, *Candida versatilis*, *Candida etchellsii*, and *Candida tropicalis*. These findings provide insight into microbial interactions and their influence on cheese texture and flavor throughout ripening. Notably, the identification of specific species—such as distinct *Candida* strains—sets this study apart from others that focus on broader taxonomic levels. Although several species commonly associated with global cheese production were detected, the use of raw cow's milk, the extended fermentation process, and the distinctive environmental conditions of Mexico contributed to a unique microbial and fungal community characteristic of this cheese. Furthermore, the detection of potentially pathogenic genera, such as *Enterobacter*, *Aeromonas*, and *Chryseobacterium*, underscores the need for continued research into their growth conditions and prevalence, as well as the development of targeted control strategies to mitigate their presence. Overall, this work represents a significant step forward in understanding the ripening process of artisanal cheeses and highlights the value of integrating metagenomic approaches with traditional culture-based methods.

## Acknowledgments

We thank Jerome Verleyen for his technical support and for granting us access to the HPC infrastructure at the Unidad Universitaria de Secuenciación Masiva y Bioinformática, Instituto de Biotecnología (UNAM), and Mabel Rodríguez for her invaluable assistance with proofreading and insightful suggestions.

## Author contributions

**Conceptualization:** Ernestina Valadez Moctezuma.

**Data curation:** Arturo Hernández Montes.

**Formal analysis:** Blanca Nayelli Ocampo Morales, Karel Estrada, Ernestina Valadez Moctezuma.

**Methodology:** Blanca Nayelli Ocampo Morales, Karel Estrada, Ernestina Valadez Moctezuma.

**Project administration:** Arturo Hernández Montes.

**Software:** Karel Estrada.

**Supervision:** Karel Estrada.

**Validation:** Karel Estrada.

**Writing – original draft:** Blanca Nayelli Ocampo Morales, Karel Estrada, Ernestina Valadez Moctezuma.

**Writing – review & editing:** Blanca Nayelli Ocampo Morales, Arturo Hernández Montes, Karel Estrada, Ernestina Valadez Moctezuma.

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
