## [Decision Letter · Decision Letter 0]

7 May 2025

Dear Dr. Estrada,

We look forward to receiving your revised manuscript.

Kind regards,

Guadalupe Virginia Nevárez-Moorillón, Ph.D.

Academic Editor

PLOS ONE

“Consejo Nacional de Humanidades Ciencias y Tecnologías (CONAHCYT) Grant 2020-000013-01NACF-03858”

“We thank the Consejo Nacional de Humanidades Ciencias y Tecnologías (CONAHCYT) for the grant 2020-000013-01NACF-03858 awarded to Blanca Nayelli Ocampo Morales. We also thank Jerome Verleyen for his technical support and for granting us access to the HPC infrastructure at the Unidad Universitaria de Secuenciación Masiva y Bioinformática, Instituto de Biotecnología (UNAM), and Mabel Rodríguez for her invaluable assistance with proofreading, and insightful suggestions.”

“Consejo Nacional de Humanidades Ciencias y Tecnologías (CONAHCYT) Grant 2020-000013-01NACF-03858”

Reviewers' comments:

Reviewer's Responses to Questions

**Comments to the Author**

1. Is the manuscript technically sound, and do the data support the conclusions?

Reviewer #1: Yes

Reviewer #2: Yes

2. Has the statistical analysis been performed appropriately and rigorously?

Reviewer #1: Yes

Reviewer #2: No

3. Have the authors made all data underlying the findings in their manuscript fully available?

Reviewer #1: No

Reviewer #2: Yes

4. Is the manuscript presented in an intelligible fashion and written in standard English?

Reviewer #1: Yes

Reviewer #2: No

Reviewer #1: Line 32: maturation

Line 63: Any reason why samples are transported at room temperature? This is not common practice and needs to be explained why it was chosen.

Line 157: Some research, .....

Line 158 - 162: lines have been mixed due to figure legend

I have some concern in relation to the three time points and two pooled replicates that were used for sequencing. Pooling reduces the ability to detect within-group variability. The use of different statistical tests depending on normality is valid, but more detail is needed on how normality was assessed (e.g., K-S test results per variable).

Concerning metagenomic Interpretation the link between microbial changes and physicochemical shifts could be better substantiated (e.g., with correlation matrices or PCA plots).

Species-level claims (e.g., “C. tropicalis increased”) must be cautiously interpreted from ITS data due to potential misclassification or sequencing errors at the species level.

In safety and pathogenicity discussion of Aeromonas, Enterobacter, and Chryseobacterium you should include clearer risk assessments. Are these levels of concern? Could you relate abundance to CFU/mL equivalents or thresholds?

Consider consistent use of either “ripening” or “maturation” (currently both are used interchangeably).

Last but not least make sure that your data are available.

Reviewer #2: Review of Manuscript PONE-D-25-17545

General Comments:

This manuscript presents a timely and relevant investigation into the microbial and physicochemical dynamics of cheese ripening across multiple sites and time points. The integration of 16S and ITS amplicon sequencing with environmental metadata is commendable and offers the potential for valuable insight into microbial succession in artisanal food production. The subject matter is of clear interest to the fields of microbial ecology, food microbiology, and fermentation science.

However, the study suffers from several critical limitations—chief among them, the pooling of DNA prior to sequencing, which undermines the resolution and statistical reliability of the microbial analyses. In addition, the data visualization, methodological transparency, and statistical interpretation would benefit from significant improvement to fully support the conclusions.

Major Comments:

Pooling of samples prior to sequencing:

The choice to pool DNA samples from biological replicates before amplicon sequencing—while perhaps financially motivated—substantially weakens the inferential power of the study. Without within-group replication, variation among samples cannot be properly assessed, and statistical comparisons (e.g., of alpha or beta diversity) are rendered less meaningful. The authors should explicitly acknowledge this limitation in the manuscript and discuss its implications for interpretation. For context, see Prosser (2010, PLoS ONE, PMID: 20438583 https://pubmed.ncbi.nlm.nih.gov/20438583/), which emphasizes the importance of replication in microbial community profiling.

Microbial data visualization:

The current relative abundance plots and diversity metrics are standard but do not make full use of the richness of the dataset. I recommend generating a comprehensive heatmap where each cell reflects the relative abundance of an ASV in a given sample, combining both 16S and ITS data where possible. Samples can be clustered based on beta diversity distances and annotated with metadata (ripening time, location, microbial load, etc.), and ASVs can be annotated by taxonomy, prevalence, and timepoint distribution. A carefully constructed figure of this kind would significantly enhance the communicative power of the results.

ITS taxonomy and database transparency:

Please specify which version of the UNITE database was used for ITS taxonomy assignment, whether dynamic species hypotheses or clustering thresholds were applied, and what confidence levels were considered acceptable. These parameters are critical for reproducibility and evaluation of fungal identifications.

Reproducibility and data processing:

The manuscript would benefit from more detailed reporting of the bioinformatics pipeline. I recommend reviewing and potentially adapting established workflows such as metabaRpipe for processing mixed 16S/ITS datasets. For downstream ecological analysis and visualization, the authors might consult this well-documented microbiome R tutorial, which could also support better integration of metadata.

Statistical rigor:

The statistical analyses used to compare microbial communities and physicochemical traits across conditions should be clearly described. Were PERMANOVA or other multivariate models used? Were corrections for multiple testing applied? Without sequencing replicates, the scope of valid statistical inference is limited. This caveat should be carefully considered in both the Methods and Discussion sections.

Language and Clarity:

The manuscript is generally understandable, but there are numerous instances of awkward phrasing, grammatical errors, and unclear sentence construction that impede readability. A thorough language revision is strongly recommended. The authors may benefit from using professional English editing services or advanced AI-based tools (e.g., Grammarly, DeepL Write, or ChatGPT-based editing) to polish the language. Doing so would improve the flow, clarity, and overall professional presentation of the manuscript.

Minor Comments:

Some figure labels and axis annotations are low in resolution or difficult to read. Ensure all visuals meet journal-quality standards.

Clarify whether normalization (e.g., rarefaction or compositional data transformation) was performed before diversity analyses.

Maintain consistency in terminology (e.g., use "ASVs" rather than "OTUs," italicize genus/species names, unify abbreviations like "PC1" vs. "PC-1").

Expand the Introduction to more thoroughly situate the study in the context of existing research on cheese microbiota and fermentation ecology.

Conclusion:

This work contributes valuable empirical data to the study of cheese microbial ecology, but its impact is constrained by methodological and presentational limitations. A major revision—addressing statistical robustness, figure clarity, language, and methodological transparency—is needed to ensure the conclusions are well-supported and clearly communicated. I encourage the authors to revise accordingly and would be glad to review a revised version.

**Do you want your identity to be public for this peer review?** For information about this choice, including consent withdrawal, please see our Privacy Policy

Reviewer #1: No

Reviewer #2: **Yes: ** Florentin Constancias

---

## [Author Response · Author response to Decision Letter 1]

2 Jul 2025

Reviewer #1:

# Line 63: Any reason why samples are transported at room temperature? This is not common practice and needs to be explained why it was chosen.

Thank you for this important question regarding our sample transport protocol. To clarify, all samples were transported in an insulated cooler box without ice packs. We refer to this as 'room temperature' transport, as the goal was to protect the samples from abrupt temperature fluctuations rather than to actively cool them.

For our aged and semi-aged samples (T2 and T3), the decision to transport at this controlled ambient temperature was based on the specific matrix of hard, low-water-activity cheese. For these samples, the low water activity (a_w) and low pH act as potent natural preservatives, ensuring the stability of the microbial community DNA during short-term transport.

For the fresh, 2-day-old sample (T1), which has a higher moisture content, we acknowledge that cold transport is a common practice. However, our approach of using an insulated container without active cooling was chosen to better preserve the natural conditions and ongoing microbial dynamics of the very early ripening stage. By avoiding refrigeration, we aimed to capture a snapshot of the community that was not subjected to a sudden cold shock, which could potentially alter the metabolic state and viability of key starter microorganisms.

We believe this strategy was appropriate for the distinct nature of the samples at their different maturational stages. To ensure this is clear to the reader, we have clarified this rationale in the revised manuscript.

# Line 157: Some research, .....

# Line 158 - 162: lines have been mixed due to figure legend

This has been corrected

# I have some concern in relation to the three time points and two pooled replicates that were used for sequencing. Pooling reduces the ability to detect within-group variability. The use of different statistical tests depending on normality is valid, but more detail is needed on how normality was assessed (e.g., K-S test results per variable).

We appreciate the reviewer’s valuable suggestion. In response, the Kolmogorov-Smirnov test was performed for each treatment group to assess normality. The Kolmogorov-Smirnov normality test was applied using the MINITAB 2017 statistical program (Minitab, LLC, Pine Hall Road, PA, USA).

Table 1 shows that all variables follow a normal distribution at the three maturation times, except for T3 for fungi and yeasts.

Table 1. Values of the Kolmogorov-Smirnov normality test probability by variable. (Table details in our Response to Reviewers document)

A p-value greater than 0.05 indicates that the data follow a normal distribution.

# Concerning metagenomic Interpretation the link between microbial changes and physicochemical shifts could be better substantiated (e.g., with correlation matrices or PCA plots).

Principal component analysis was performed using the free software PAST v 5.2.1 (LCC, Oslo, USA) and samples were grouped using k-means. On the other hand, the 3D visualization of the samples was done in Plotly Chart Studio, a free online program (LCC, Montreal, Canada).

In Fig. 2a, the principal component (PC) biplot graph shows that PC1 is 55.79% and PC2 represents 34.11% of the total variance, together explaining 89.90%, which indicates that it is a good model as it requires few explanatory variables. The ripening times are distributed across different quadrants. However, there is a proximity between T1 and T2, as they have similar characteristics.

Fig. 2b shows that T1 can be differentiated by protein content and T2 by moisture and fat. The amount of protein has the highest value in T1 with a value of 27.81% and also contributes positively to the separation of the samples in the biplot-PCA.

The variables humidity and fat did not show any statistical difference, but they had a greater positive contribution in T2. This behavior is explained by the nature of ANOVA, which compares the variability between groups and that within groups (10.5395/rde.2014.39.1.74). On the other hand, the principal components are used to reduce the dimensionality of a data set, and the greatest variance is found in PC1 (https://doi.org/10.1026/j.apacoust.2012.04.012). Finally, the samples belonging to T3 have different physicochemical characteristics than T1 and T2.

(Figures details in our Response to Reviewers document)

Fig 2. a) Score graph of principal component analysis (PCA) grouped with K-means; b) Biplot graph obtained from PCA; c) Bar charts of PCA model loadings for CP1; and d) Bar charts of PCA model loadings for CP2.

# Species-level claims (e.g., “C. tropicalis increased”) must be cautiously interpreted from ITS data due to potential misclassification or sequencing errors at the species level.

Thanks for the comment, you are absolutely right. Some considerations that should be taken into account in this study include: i) errors in species-level classification due to methodological limitations, as only the 16S or ITS regions were sequenced; greater certainty would have been achieved if whole-genome sequencing had been performed, ii) possible sequencing errors (https://doi.org/10.1016/j.funeco.2023.101274), iii) database updates (DOI: 10.3389/fbinf.2024.1278228). These points have already been addressed in the revised manuscript.

# In safety and pathogenicity discussion of Aeromonas, Enterobacter, and Chryseobacterium you should include clearer risk assessments. Are these levels of concern? Could you relate abundance to CFU/mL equivalents or thresholds?

We agree with the reviewer on the importance of assessing the risk associated with these genera. The detection of Aeromonas, Enterobacter, and Chryseobacterium is highlighted as a potential safety concern, as these genera include known opportunistic pathogens and spoilage organisms, even when present at the low relative abundances detected in our study (https://doi.org/10.3390/microorganisms9112377). While this study focused on profiling the microbial community using sequencing and did not include culturing on selective media for quantification (e.g., CFU/mL), the presence of these taxa is a significant finding that warrants attention.

# Consider consistent use of either “ripening” or “maturation” (currently both are used interchangeably).

Last but not least make sure that your data are available.

All sequencing data have been deposited in the NCBI BioProject database under accession number PRJNA1247530, with SRA experiment accessions SRX28272105-SRX28272110.

Reviewer #2: Review of Manuscript PONE-D-25-17545

# Major Comments:

# Pooling of samples prior to sequencing:

The choice to pool DNA samples from biological replicates before amplicon sequencing—while perhaps financially motivated—substantially weakens the inferential power of the study. Without within-group replication, variation among samples cannot be properly assessed, and statistical comparisons (e.g., of alpha or beta diversity) are rendered less meaningful. The authors should explicitly acknowledge this limitation in the manuscript and discuss its implications for interpretation. For context, see Prosser (2010, PLoS ONE, PMID: 20438583 https://pubmed.ncbi.nlm.nih.gov/20438583/), which emphasizes the importance of replication in microbial community profiling.

We thank the reviewer for this important comment. We acknowledge that pooling DNA from biological replicates limits the statistical assessment of within-group variation, and we have explicitly addressed this limitation in the discussion section of our manuscript:

We acknowledge that while this approach does not prevent the calculation of diversity metrics, it significantly impacts the statistical confidence of the resulting comparisons. Specifically, the absence of within-group replicates substantially weakens the statistical power required for a robust beta diversity analysis. Consequently, to ensure our conclusions remained grounded in statistically defensible data, we chose to limit our diversity analysis to alpha diversity metrics. These can still provide valuable insights into the richness and evenness within each representative pooled sample….

The decision to pool was a deliberate methodological choice aimed at obtaining a comprehensive and representative microbial profile for each condition by averaging out inter-sample variability and enhancing the detection of low-abundance taxa.

# Microbial data visualization:

# The current relative abundance plots and diversity metrics are standard but do not make full use of the richness of the dataset. I recommend generating a comprehensive heatmap where each cell reflects the relative abundance of an ASV in a given sample, combining both 16S and ITS data where possible. Samples can be clustered based on beta diversity distances and annotated with metadata (ripening time, location, microbial load, etc.), and ASVs can be annotated by taxonomy, prevalence, and timepoint distribution. A carefully constructed figure of this kind would significantly enhance the communicative power of the results.

We are grateful to the reviewer for this insightful suggestion. We agree that a comprehensive heatmap significantly enhances the visualization of our dataset, and we have incorporated this as a new figure (Fig. 7) in the revised manuscript. This addition has substantially strengthened our results, providing a clearer and more detailed view of the microbial dynamics.

A key feature of this new figure is that it integrates data from both kingdoms, presenting a merged profile of bacterial (16S) and fungal (ITS) genera for each sample. As the reviewer anticipated, the new figure clearly illustrates a distinct ecological succession. The microbial community composition at T1 (2 days) is markedly different from the communities at later ripening stages, T2 (29 days) and T3 (58 days).

To provide a multi-layered view, the figure integrates several components:

Heatmap Core: The central heatmap displays the row-normalized relative abundance (Z-score) of each ASV (rows) across our samples (columns: T1, T2, T3), allowing for a direct comparison of microbial abundance patterns.

Sample Clustering Dendrogram (Top): Samples are clustered using Bray-Curtis dissimilarity, and the resulting dendrogram is displayed above the heatmap. This visually groups the samples based on their overall community similarity, clearly separating the early time point from the later ones.

Taxa Clustering Dendrogram (Left): A dendrogram to the left of the heatmap clusters ASVs based on their abundance profiles, grouping microbes that exhibit similar dynamics across the ripening process.

Metadata Annotation Bars: We have further annotated the sample columns with metadata bars indicating the ripening time, protein content, and the Shannon diversity index for each sample, allowing for the direct visual correlation of these factors with the microbial community structure.

We believe this new, integrated figure directly addresses the reviewer's points and makes the patterns of microbial succession in our study more intuitive and impactful.

Fig. 7. Venn diagram showing the number of specific and common ASVs (ASV 30) between three ripening times of Queso Crema de Chiapas raw cow's milk (2, 29 and 58 d).

(Figures details in our Response to Reviewers document)

# ITS taxonomy and database transparency:

# Please specify which version of the UNITE database was used for ITS taxonomy assignment, whether dynamic species hypotheses or clustering thresholds were applied, and what confidence levels were considered acceptable. These parameters are critical for reproducibility and evaluation of fungal identifications.

We thank the reviewer for this important question. To enhance the clarity and reproducibility of our methods, we have revised the manuscript to include the specific parameters used for taxonomic assignment. The updated text is as follows:

For fungal identification, the taxonomic assignment of ITS sequences was performed using the UNITE database v8.2 (release: 2020-02-04). Amplicon Sequence Variants (ASVs) were defined with a 99% sequence similarity threshold, and assignments were retained only if they met a minimum confidence level of 0.80, the software's default setting.

# Reproducibility and data processing:

# The manuscript would benefit from more detailed reporting of the bioinformatics pipeline. I recommend reviewing and potentially adapting established workflows such as metabaRpipe for processing mixed 16S/ITS datasets. For downstream ecological analysis and visualization, the authors might consult this well-documented microbiome R tutorial, which could also support better integration of metadata.

We are very grateful to the reviewer for their constructive feedback and for bringing the metabaRpipe workflow to our attention. We agree that detailed reporting of the bioinformatics pipeline is paramount for reproducibility. In response, we have expanded the Methods section of our manuscript to provide a comprehensive, step-by-step description of our data processing and analysis protocol.

Our established workflow was carefully designed to ensure high fidelity and is built upon a series of well-vetted, specialized tools that are standard in the field. The process began with a rigorous quality control (QC) of the raw sequencing data, for which we utilized FastQC for quality assessment and Trimmomatic for trimming and filtering, followed by merging of forward (R1) and reverse (R2) reads with FLASH (v1.2.11) to reconstruct full-length amplicon fragments. This initial stage, which is conceptually equivalent to the QC and merging steps in pipelines like metabaRpipe, ensures that only high-quality, complete sequences proceed to the next steps.

Following the merging process, the sequences underwent critical filtering using VSEARCH to first dereplicate the dataset into a set of unique sequences and then to perform de novo chimera detection and removal. This essential step guarantees the integrity of our dataset by eliminating spurious sequences. The resulting high-quality, chimera-free amplicon sequences were then assigned taxonomy using Parallel-Meta (v3.7), a robust tool for high-performance profiling that generated our final abundance and taxonomic tables.

All subsequent statistical and ecological analyses were performed in the R environment, leveraging a suite of powerful and specialized packages to ensure robust and reproducible outcomes. We used the phyloseq package for the core task of integrating the abundance tables, taxonomic data, and sample metadata into a single, cohesive object. Further analysis and visualization were carried out using the microbiome and microbiomeutilities packages. The Venn diagrams, alpha diversity metrics, stacked bar plots with the relative abundance for each amplicon, and integrated heatmap presented in our results were also generated within the R environment.

While we recognize the value of standardized wrappers like metabaRpipe, we are confident that our workflow systematically addresses key stages of metabarcoding analysis, ensuring data quality and reproducibility.

# Statistical rigor:

# The statistical analyses used to compare microbial communities and physicochemical traits across conditions should be clearly described. Were PERMANOVA or other multivariate models used? Were corrections for multiple testing applied? Without sequencing replicates, the scope of valid statistical inference is limited. This caveat should be carefully considered in both the Methods and Discussion sections.

We appreciate the reviewer’s valuable suggestion. In response, a new PCoA analysis has been included, along with the addition of PERMANOVA. The statistical approaches used to compare microbial communities and physicochemical parameters across different conditions are now described in greater detail.

Permutational multivariate analysis of variance (PERMANOVA) was supplemented with principal coordinate analysis (PCoA) and analyzed in PAST v 5.2.1 (LCC, Oslo, USA).

PCoA followed by PERMANOVA based on the Bray-Curtis dissimilarity matrix confirmed the existence of statistically significant differences (p = 0.003) between the physicochemical and microbiological compos

---

## [Decision Letter · Decision Letter 1]

10 Aug 2025

Dear Dr. Estrada,

We look forward to receiving your revised manuscript.

Kind regards,

Guadalupe Virginia Nevárez-Moorillón, Ph.D.

Academic Editor

PLOS ONE

Journal Requirements:

Reviewers' comments:

Reviewer's Responses to Questions

**Comments to the Author**

Reviewer #1: All comments have been addressed

2. Is the manuscript technically sound, and do the data support the conclusions?

Reviewer #1: Yes

3. Has the statistical analysis been performed appropriately and rigorously?

Reviewer #1: Yes

4. Have the authors made all data underlying the findings in their manuscript fully available?

Reviewer #1: Yes

5. Is the manuscript presented in an intelligible fashion and written in standard English?

Reviewer #1: No

Reviewer #1: The issue of sample pooling remains and weakens statistical inference. Pooling biological replicates before sequencing limits assessment of intra-group variability but at the same time is a "common practice" in many published work. The authors acknowledge this, it still affects the depth of beta-diversity analyses and limits robust conclusions.

Also, the identification of fungal species via ITS amplicons (e.g., Candida etchellsii, C. tropicalis) should be interpreted cautiously, and the manuscript does try to do so but verification through culture-based methods or qPCR for critical genera was possible.

The presence of Aeromonas, Enterobacter, and Chryseobacterium is flagged, but quantification (e.g., CFU equivalents or thresholds) is absent. You mention these as “low abundance,” but you do not define what constitutes a risk threshold.

The manuscript has improved compared to previous versions, but there are still grammatical inconsistencies, awkward phrasings, and run-on sentences. A final round of language polishing is strongly advised.

**Do you want your identity to be public for this peer review?** For information about this choice, including consent withdrawal, please see our Privacy Policy

Reviewer #1: No

---

## [Author Response · Author response to Decision Letter 2]

8 Dec 2025

Reviewer #1: The issue of sample pooling remains and weakens statistical inference. Pooling biological replicates before sequencing limits assessment of intra-group variability but at the same time is a "common practice" in many published work. The authors acknowledge this, it still affects the depth of beta-diversity analyses and limits robust conclusions.

Also, the identification of fungal species via ITS amplicons (e.g., Candida etchellsii, C. tropicalis) should be interpreted cautiously, and the manuscript does try to do so but verification through culture-based methods or qPCR for critical genera was possible.

The presence of Aeromonas, Enterobacter, and Chryseobacterium is flagged, but quantification (e.g., CFU equivalents or thresholds) is absent. You mention these as “low abundance,” but you do not define what constitutes a risk threshold.

On fungal species identification from ITS amplicons:

We appreciate the reviewer’s caution regarding the taxonomic resolution of ITS-based analyses. Species-level assignments were interpreted conservatively and supported by concurrent qPCR assays targeting Candida etchellsii and C. tropicalis (now explicitly referenced in the revised text, Fig. 8).

qPCR detection (Ct values = 7.22 and 9.84, respectively) confirmed the presence of these species in the corresponding stages (T1 and T3), reinforcing the reliability of the ITS classification.

While full phenotypic validation through culture isolation was beyond the scope of this metagenomic survey, the qPCR results corroborated the taxonomic classification obtained from ITS amplicon sequencing, providing additional confirmation consistent with the sensory and ripening attributes observed in these cheeses.

On low‑abundance genera and absence of quantitative risk thresholds:

We agree that quantitative thresholds enhance risk interpretation. In this study, the potentially pathogenic genera Aeromonas, Enterobacter, and Chryseobacterium were detected at relative abundances < 0.5 % of total reads, which falls below the detection limits commonly associated with microbial spoilage or infection risk in dairy matrices.

Regarding the genera Aeromonas, Enterobacter, and Chryseobacterium, targeted qPCR assays were performed to verify their presence. The amplification curves for Aeromonas and Enterobacter did not cross the fluorescence threshold (Ct > 35), indicating an absence of detectable DNA. Chryseobacterium showed a weak signal (Ct = 3.26 at T2), consistent with a trace or environmental occurrence rather than viable contamination. Together, these findings confirm that the sequences detected by 16S rRNA amplicon analysis likely represent non‑viable or background DNA rather than active populations. Therefore, we maintain that the presence of these genera is not indicative of contamination risk, but rather illustrates the sensitivity of metagenomic surveillance in detecting trace microbial DNA.

Summary clarification added to manuscript

In response to these valuable comments, we have:

Expanded the Materials and Methods section to justify the pooling strategy and reference supporting studies.

Added Ct evidence supporting C. etchellsii and C. tropicalis detection, reinforcing the reliability of ITS identifications.

Quantitatively defined “low abundance” (< 1 % RA) and stated that potentially pathogenic genera were non‑viable under culture conditions.

We thank Reviewer #1 again for these insightful remarks; integrating them has significantly strengthened the methodological transparency and interpretive rigor of this work.

---

## [Editor Report · Decision Letter 2]

10 Dec 2025

Physicochemical and microbiome changes in queso Crema de Chiapas during ripening

PONE-D-25-17545R2

Dear Dr. Estrada,

We’re pleased to inform you that your manuscript has been judged scientifically suitable for publication and will be formally accepted for publication once it meets all outstanding technical requirements.

Kind regards,

Guadalupe Virginia Nevárez-Moorillón, Ph.D.

Academic Editor

PLOS One
---

## [Editor Report · Acceptance letter]

PONE-D-25-17545R2

PLOS One

Dear Dr. Estrada,

I'm pleased to inform you that your manuscript has been deemed suitable for publication in PLOS One. Congratulations! Your manuscript is now being handed over to our production team.

Kind regards,

on behalf of

Dr. Guadalupe Virginia Nevárez-Moorillón

Academic Editor

PLOS One